# DeMo: Decoupled Momentum Optimization

**Bowen Peng**[†]      **Lizhang Chen**[*‡]      **Baiyu Su**[*‡]

**Jeffrey Quesnelle**[†]      **Diederik P. Kingma**[§]      **Qiang Liu**[‡]

## Abstract

Scaling neural network training increasingly depends on synchronous data-parallelism, yet full-precision gradient all-reduce imposes a severe communication bottleneck. We propose *Decoupled Momentum Optimization* (DeMo), a drop-in replacement for any momentum-based optimizers that significantly reduces the communication bandwidth while maintaining convergence. DeMo (i) decouples local momentum updates, (ii) applies a fast orthonormal transform (e.g., DCT) followed by top-$k$ sparsification, and (iii) reuses the momentum buffer as error feedback via momentum subtraction. This design reduces per-step communication by up to two orders of magnitude with minimal computational overhead. Experiments on 300M- and 1B-parameter DeMo language models show DeMo transmits up to 85× less data per GPU than AdamW-DDP while achieving comparable loss and accuracy. DeMo is topology-agnostic and enables training across multi-datacenter or Ethernet-based setups. Code is available at `https://github.com/bloc97/DeMo`.

## 1 Introduction

The proliferation of large-scale foundation models, with parameter counts reaching the billions (Achiam et al., 2023; Grattafiori et al., 2024; Liu et al., 2024a), has been transformative across numerous domains. Training these massive models within a tractable timeframe necessitates distributing the heavy computational workload across a large number of accelerators (Duan et al., 2024). This is typically managed through a hierarchy of parallelism strategies, such as Distributed Data Parallelism (DDP), Pipeline Parallelism (PP), and Fully Sharded Data Parallelism (FSDP) (Zhao et al., 2023).

Among these, distributed data parallelism remains the most widely used strategy but introduces a major communication bottleneck. In standard DDP, each worker synchronizes its locally computed gradients with all others before every optimization step—typically via an `All-Reduce` operation. The communication cost is directly proportional to the model's size, reaching terabytes per step for state-of-the-art models. This overhead demands expensive, high-bandwidth interconnects (e.g., NVLink, InfiniBand) and geographically co-located clusters Wei et al. (2024), driving up costs and limiting scalability. Alleviating this communication burden is thus critical for enabling more efficient and accessible large-scale training Cao et al. (2023).

In this paper, we show that the gradient information exchanged during distributed training is highly redundant. Our key insight is that this redundancy can be managed more effectively by communicating compressed momentum updates in a transformed space rather than transmitting raw gradients. Prior sparsification approaches directly applied to gradients incur sparse update patterns and often harm performance Lin et al. (2018b); Stich et al. (2018); Aji and Heafield (2017); in contrast, our blockwise transform and top-$k$ sparsification preserve accuracy by operating in the transformed domain. Moreover, the momentum buffer itself naturally serves as an implicit error accumulator, eliminating the need for explicit error-feedback mechanisms that introduce substantial GPU memory overhead (Karimireddy et al., 2019). Building on these observations, we propose **DeMo** (**De**coupled **Mo**mentum Optimization), a general-purpose distributed optimization framework compatible with

---

[*]Equal contribution.

[‡]University of Texas at Austin

[§]This work was done in the author's personal capacity before joining Anthropic.

[†]Nous Research. Correspondence: {`bloc,emozilla`}@nousresearch.com

common momentum-based optimizers, including SGD with momentum (Sutskever et al., 2013), Lion (Chen et al., 2023), and Muon (Jordan et al., 2024). DeMo fundamentally rethinks synchronization: instead of exchanging dense gradients, each worker communicates a compressed version of its local momentum, subtracts the decoded update from its buffer to track uncommunicated information, and then applies the aggregated momentum to update the parameters. This simple but powerful design drastically reduces communication while retaining the benefits of momentum-based optimization.

This approach drastically reduces the required inter-accelerator communication bandwidth, potentially by several orders of magnitude. Our primary contributions are:

- We identify the momentum term in distributed optimization as a compressible, information-rich surrogate for raw gradients.

- We propose **DeMo**, a communication-efficient framework that decouples momentum updates and leverages the momentum itself as a built-in error feedback mechanism.

- We demonstrate that our method significantly relaxes the hardware and co-location constraints for large-scale training, paving the way for more flexible, geographically distributed, and cost-effective training paradigms.

The remainder of this paper is organized as follows. We review related work in Section 1, 4. In Section 2, we present the DeMo algorithm and its theoretical guarantees. Section 3 provides empirical results demonstrating DeMo's effectiveness, followed by ablation studies. Finally, we conclude in Section 6.

## 2 DeMo: Decoupled Momentum Optimization

**Problem setting.** We consider the standard stochastic optimization problem

$$\min_{\boldsymbol{X} \in \mathbb{X}} \mathcal{L}(\boldsymbol{X}) := \mathbb{E}_{\xi \sim \mathcal{D}} \left[ \mathcal{L}(\boldsymbol{X}, \xi) \right], \tag{1}$$

where $\boldsymbol{X} \in \mathbb{X}$ is (possibly a series of) parameter tensors to be optimized, $\mathcal{L}(\boldsymbol{X}, \xi)$ is a sample loss function (e.g., cross-entropy), and the expectation is taken over data samples $\xi$ drawn from distribution $\mathcal{D}$. The optimal value is denoted by $\mathcal{L}^* := \inf_{\boldsymbol{X} \in \mathbb{X}} \mathcal{L}(\boldsymbol{X})$, which we assume is finite and bounded from below. We operate in a distributed data-parallel setting with $I$ workers, each holding synchronized copies of parameters $\boldsymbol{X}$. At each step $t$, worker $i$ computes a stochastic gradient $G_t^i = \frac{1}{n_{\text{batch}}} \sum_{n=1}^{n_{\text{batch}}} \nabla \mathcal{L}(\boldsymbol{X}_t, \xi_n^i)$ using a per-worker microbatch of size $n_{\text{batch}}$.

### 2.1 Algorithm

We start to build DeMo from the standard DDP optimization pipeline and common momentum-based optimizers, such as SGD with momentum (Sutskever et al., 2013), Signum (Bernstein et al., 2018), and Muon (Jordan et al., 2024). Our proposed method introduces three key modifications to reduce communication overhead and improve efficiency: (1) decoupled local momentum updates to be communicated instead of gradients, (2) structured tensor compression, and (3) momentum subtraction as error feedback.

**Decoupled Local Momentum Updates.** Standard DDP pipelines typically synchronize gradients across workers immediately after every local gradient tensor being computed. In contrast, we remove the global `all_reduce` synchronization of micro-batch gradients $G_t^i$, allowing local momentum buffers $M_t^i$ on each worker to evolve independently:

$$\boldsymbol{M}_t^i = \beta \boldsymbol{M}_{t-1}^i + (1 - \beta) \boldsymbol{G}_t^i, \tag{2}$$

where $\beta$ is the momentum coefficient. Aggregating gradients or momenta is theoretically equivalent due to linearity; however, directly synchronizing dense momentum tensors is communication-intensive. Thus, we propose a structured tensor compression pipeline to significantly reduce this overhead.

---

**Algorithm 1** DeMo: Decoupled Momentum

---

**Require:** learning rate $\eta$, momentum coefficient $\beta$, weight decay $\lambda$, sparsity budget $k$, DCT matrices $\{\boldsymbol{P}_j\}_{j=0}^{d-1}$

---

1: **Initialization:** parameters $\mathbf{X}_0$ and global momentum $\mathbf{M}_0 \leftarrow \mathbf{0}$
2: **for** $t = 1, 2, \ldots$ **do**
3:      **# on each worker** $i \in \{0, \ldots, N-1\}$
4:      $\mathbf{G}_t^i \leftarrow \nabla \mathcal{L}(\mathbf{X}_{t-1}; \xi_t^i)$                             # local stochastic gradient
5:      $\mathbf{M}_t^i \leftarrow \beta \mathbf{M}_{t-1}^i + \mathbf{G}_t^i$                           # update local momentum buffer
6:      **for** each chunk $\mathbf{M}_t^{i,[\ell]}$ of $\mathbf{M}_t^i$ **do**
7:          $\mathbf{Q}_t^{i,[\ell]} \leftarrow \text{Top-}k\big(\text{DCT}(\mathbf{M}_t^{i,[\ell]}; \{\boldsymbol{P}_j\}), k\big)$      # blockwise transform & sparsify in the $\ell$-th chunk
8:          $\mathbf{M}_t^{i,[\ell]} \leftarrow \mathbf{M}_t^{i,[\ell]} - \text{IDCT}(\mathbf{Q}_t^{i,[\ell]}; \{\boldsymbol{P}_j^\top\})$      # in-place residual in momentum buffer
9:      **end for**
10:     **send** $\{\mathbf{Q}_t^{i,[\ell]}\}$ **to server**
11:     **# on the parameter server**
12:     **for** each chunk $\{\mathbf{Q}_t^{i,[\ell]}\}$ **do**
13:        $\mathbf{M}_t^{[\ell]} \leftarrow \text{IDCT}\Big(\sum_i \mathbf{Q}_t^{i,[\ell]}; \{\boldsymbol{P}_j^\top\}\Big)$    # aggregate sparse updates and reconstruct momentum chunk
14:     **end for**
15:     $\mathbf{X}_t \leftarrow \mathbf{X}_{t-1} - \eta\big(\text{sgn}(\mathbf{M}_t) + \lambda \mathbf{X}_{t-1}\big)$
16:     **broadcast** $\text{sgn}(\mathbf{M}_t)$ **to all workers**
17: **end for**

---

**Structured Tensor Compression.** Our compression pipeline comprises three sequential steps: tensor chunking, blockwise linear projection, and top-$k$ sparsification.

*Tensor Chunking.* Given a momentum tensor $M \in \mathbb{R}^{n_0 \times \cdots \times n_{d-1}}$, we factorize each dimension as $n_i = c_i s_i$ with chunk number $c_i$ and size $s_i$ to partition $M$ into smaller blocks:

$$\mathcal{B}(M) = \{\boldsymbol{B}_\mathbf{k} \mid \mathbf{k} \in [c_0] \times \cdots \times [c_{d-1}]\},$$

where each block $\boldsymbol{B}_\mathbf{k} \in \mathbb{R}^{s_0 \times \cdots \times s_{d-1}}$. *Blockwise Linear Projection.* After chunking, each block $\boldsymbol{B}_\mathbf{k} \in \mathcal{B}(M)$ undergoes a separable multilinear transformation:

$$\boldsymbol{Q}_\mathbf{k} = \mathcal{T}(\boldsymbol{B}_\mathbf{k}; \boldsymbol{P}_0, \ldots, \boldsymbol{P}_{d-1}), \quad \boldsymbol{P}_i \in \mathbb{R}^{s_i \times s_i}, \tag{3}$$

where $\mathcal{T}$ multiplies block $\boldsymbol{B}_\mathbf{k}$ by projection matrices $\boldsymbol{P}_i$ along each tensor dimension. For $d = 2$, this reduces to a simple bilinear form of matrix multiplication $\boldsymbol{Q}_\mathbf{k} = \boldsymbol{P}_0 \boldsymbol{B}_\mathbf{k} \boldsymbol{P}_1^\top$.

In practice, we consider two types of projection bases: random orthonormal matrices, sampled freshly at each step (shared across all workers), and the Discrete Cosine Transform (DCT). Our experiments in Section 3 demonstrate that projections clearly outperform no projection ($\boldsymbol{P}_i = \boldsymbol{I}$). While random projections perform marginally better, DCT is computationally more efficient due to its fast implementation with Fast Fourier Transform and precomputed only once before training. Thus, we default to DCT in all experiments apart from ablations.

*Top-$k$ Sparsification.* Following projection, in order to reduce communication, each worker prepares only the top-$k$ largest-magnitude coefficients per block for communication:

$$\hat{Q}_\mathbf{k} = \text{Top-k}(\boldsymbol{Q}_\mathbf{k}, k) \quad \text{such that} \quad \|\hat{Q}_\mathbf{k}\|_0 = k. \tag{4}$$

Thereby the uploading bandwidth is reduced by $(\prod_i s_i)/k$ times and for small $k$ this can be significant. These sparse blocks are communicated using an `All_Gather` operation and averaged over all workers to form:

$$\overline{\boldsymbol{Q}}_\mathbf{k} = \text{AvgAllGather}(\hat{Q}_\mathbf{k}). \tag{5}$$

**Momentum Reconstruction and Parameter Update.** After communication, each worker reconstructs the momentum tensor from the aggregated sparse blocks via inverse projection and unchunking by concatenation:

$$M_t^* = \mathcal{B}^{-1}\big\{\mathcal{T}^{-1}\big(\overline{\boldsymbol{Q}}_\mathbf{k}; \boldsymbol{P}_0^{-1}, \ldots, \boldsymbol{P}_{d-1}^{-1}\big)\big\}, \tag{6}$$

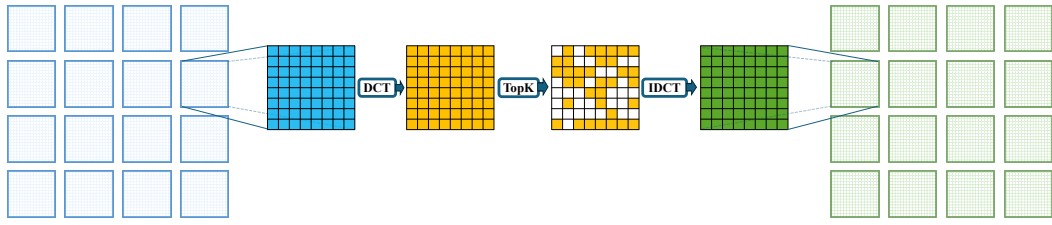

Figure 1: Chunk-wise Discrete Cosine Transform (DCT), top-$k$ coefficient selection, and inverse DCT reconstruction.

where $\boldsymbol{P}_i^{-1}$ denotes inverse projection matrices. To enhance convergence during neural network training, we apply a transformation $\phi(\cdot)$ to the reconstructed momentum $M_t^*$, chosen according to the base optimizer: $\phi(\boldsymbol{M}) = \boldsymbol{M}$ for SGD, $\phi(\boldsymbol{M}) = \text{sign}(\boldsymbol{M})$ (elementwise) for Signum, and $\phi(\boldsymbol{M}) = \boldsymbol{M}(\boldsymbol{M}^\top \boldsymbol{M} + \epsilon \boldsymbol{I})^{-1/2}$ for Muon. The final parameter update is given by

$$\boldsymbol{X}_{t+1} = \boldsymbol{X}_t - \eta_t(\phi(\boldsymbol{M}_t^*) + \lambda \boldsymbol{X}_t). \tag{7}$$

Algorithm 4 summarizes this entire procedure.

**Momentum Subtraction as Error Feedback.**  Inspired by error feedback mechanisms used with sparsification techniques, we propose updating the decoupled momentum buffer $M_t^i$ via momentum subtraction:

$$\boldsymbol{M}_t^i \leftarrow \boldsymbol{M}_t^i - \alpha\,\mathcal{B}^{-1}\big\{\mathcal{T}^{-1}(\boldsymbol{Q_k}; \boldsymbol{P}_0^{-1}, \ldots, \boldsymbol{P}_{d-1}^{-1})\big\}, \tag{8}$$

where $\alpha \in (0, 1]$ is the momentum subtraction coefficient. Unlike traditional error-feedback schemes, which require additional memory storage, our method naturally reuses the momentum buffer as an accumulator of uncommunicated information. This subtraction ensures each iteration communicates novel information, accumulating previously omitted updates over subsequent steps and promoting convergence. The decay of past gradient information is controlled by both $\alpha$ and $\beta$.

**Complexity Analysis.**  Without chunking, projecting a momentum matrix of size $N \times N$ requires $\mathcal{O}(N^3)$ computation and $\mathcal{O}(N^2)$ storage. By partitioning the matrix into $C^2$ blocks of size $(N/C) \times (N/C)$, computational complexity reduces linearly to $\mathcal{O}(N^3/C)$, and memory usage quadratically to $\mathcal{O}(N^2/C^2)$. Since all momentum tensors share the same set of projection matrices $\{P_i\}_{i=0}^{d-1}$, the additional memory overhead remains constant and negligible compared to the overall memory requirements.

## 2.2 Theoretical Analysis

To analyze the convergence properties of the DeMo algorithm, we adopt standard assumptions commonly used in stochastic optimization Bernstein and Newhouse (2024); Défossez et al. (2022); Liu et al. (2024b).

**Assumption 1** (Variance). *Let samples $\xi \sim \mathcal{D}$ be i.i.d., and define the expected gradient as $\nabla\mathcal{L}(\boldsymbol{X}) := \mathbb{E}[\nabla\mathcal{L}(\boldsymbol{X}, \xi)]$. The stochastic gradient estimator at each worker is given by*

$$G^i(\boldsymbol{X}) = \frac{1}{n_{\text{batch}}} \sum_{n=1}^{n_{\text{batch}}} \nabla\mathcal{L}(\boldsymbol{X}, \xi_n),$$

*and satisfies the bounded variance condition:*

$$\mathbb{E}\left[\left\|G^i(\boldsymbol{X}) - \nabla\mathcal{L}(\boldsymbol{X})\right\|_{\text{F}}^2\right] \leq \frac{\sigma^2}{n_{\text{batch}}}, \quad \forall i \in [N],\ \boldsymbol{X} \in \mathbb{X},$$

*where $\sigma^2 > 0$ is a finite constant and $n_{\text{batch}}$ is the mini-batch size.*

**Assumption 2** (L-Smoothness). *The objective function $\mathcal{L}(\boldsymbol{X})$ is differentiable and $L$-smooth; that is, for all $\boldsymbol{X}, \boldsymbol{Y} \in \mathbb{X}$,*

$$\|\nabla\mathcal{L}(\boldsymbol{Y}) - \nabla\mathcal{L}(\boldsymbol{X})\|_{\text{F}} \leq L\,\|\boldsymbol{Y} - \boldsymbol{X}\|_{\text{F}}.$$

*Equivalently, for all $\boldsymbol{X}, \boldsymbol{Y} \in \mathbb{X}$,*

$$\mathcal{L}(\boldsymbol{Y}) \leq \mathcal{L}(\boldsymbol{X}) + \langle\nabla\mathcal{L}(\boldsymbol{X}), \boldsymbol{Y} - \boldsymbol{X}\rangle + \frac{L}{2}\|\boldsymbol{Y} - \boldsymbol{X}\|_{\text{F}}^2.$$

**Assumption 3** (Bounded Gradient). *For any $\boldsymbol{X} \in \mathbb{X}$ and $\xi \sim \mathcal{D}$, the stochastic gradient satisfies*

$$\|\nabla\mathcal{L}(\boldsymbol{X};\xi)\|_1 \leq R,$$

*for some constant $R > 0$.*

We now present the convergence guarantee for the proposed Decoupled Momentum Optimization (DeMo) algorithm under the above assumptions.

**Theorem 1** (Convergence of DeMo). *Under Assumptions 1, 2, and 3, the sequence $\{\boldsymbol{X}_t\}_{t=1}^{T}$ from Algorithm 4 satisfies:*

$$\frac{1}{T}\sum_{t=1}^{T}\mathbb{E}\left[\|\nabla\mathcal{L}(\boldsymbol{X}_t)\|_1\right] \leq \frac{\mathbb{E}[\mathcal{L}(\boldsymbol{X}_0) - \mathcal{L}(\boldsymbol{X}_T)]}{T\eta} + 2LD\eta + \frac{\sigma}{\sqrt{Nn_{\text{batch}}}}$$

$$+ R\sqrt{D}\frac{\sqrt{1-\frac{k}{M}}}{1-\beta\sqrt{1-\frac{k}{M}}}\left(\beta + 2D\sqrt{\frac{2\pi}{N}}\left(1 + \sqrt{\frac{M}{k}}\right)\right),$$

with the choice of step size $\eta = \Theta\left(\frac{1}{\sqrt{T}}\right)$ and momentum $\beta = O\left(\frac{1}{\sqrt{T}}\right)$, the convergence rate is: $O\left(\frac{1}{\sqrt{T}}\right) + O\left(\frac{1}{\sqrt{N}}\right)$.

# 3 EXPERIMENTAL RESULTS

We evaluate our proposed Decoupled Momentum Optimization (DeMo) algorithm in the context of large-scale language model pretraining, where communication efficiency is of uttermost concern. All experiments are conducted using the OLMo (Groeneveld et al., 2024) framework, a reproducible and scalable platform for training open-weight language models. The complete implementation of DeMo, along with all configuration files necessary to reproduce our results, is available at:[1].

## 3.1 EXPERIMENTAL SETUP

**Models and Baselines.** We evaluate DeMo on Transformer-based decoder-only language models at two scales. **OLMo-300M** contains 320 million non-embedding parameters, and **OLMo-1B** contains 1.18 billion non-embedding parameters Full model specifications are provided in the Appendix.

We compare DeMo against the standard AdamW optimizer using default hyperparameters recommended by OLMo: $\beta_1 = 0.9$, and weight decay $\lambda = 0.1$. The learning rate schedules use linear warmup followed by cosine decay, scaled proportionally to the total training steps. For the runs in Figure 3.1 (and consequently Figure 4), we adopt the best recommended learning rates from Groeneveld et al. (2024), since tuning over the full training run would incur infeasible computational costs. We tune the AdamW second-order momentum $\beta_2$ over the set $[0.95, 0.98, 0.99, 0.999]$ using a reduced training budget of 1B tokens. Results with extensively searched hyperparameters for all optimizers are provided in Figure 5. For all DeMo experiments presented in the main body of the paper, we default to use chunk size of $s = 64$ and therefore making $64 \times 64$ chunks for matrix tensors in the transformers and $64$ sized chunks for vector tensors like layernorms. We found setting a larger coefficient of $\beta = 0.999$ significantly helps with convergence comparing to standard value of $0.9$ when the momentum subtraction is turned on and we therefore default to this value over all experiments. We left the momentum coefficient to be a tunable parameter from the set of $\{0.2, 0.5, 1.0\}$.

**Datasets.** All models are pretrained on the Dolma v1.5 corpus. To study optimizer behavior near convergence, we train both OLMo-300M and OLMo-1B on 100 billion tokens—significantly beyond the Chinchilla token budget of 20 tokens per parameter. Due to computational constraints, all ablation studies are conducted using the 300M model trained under the 20 tokens-per-parameter rule.

---

[1]https://github.com/bloc97/DeMo

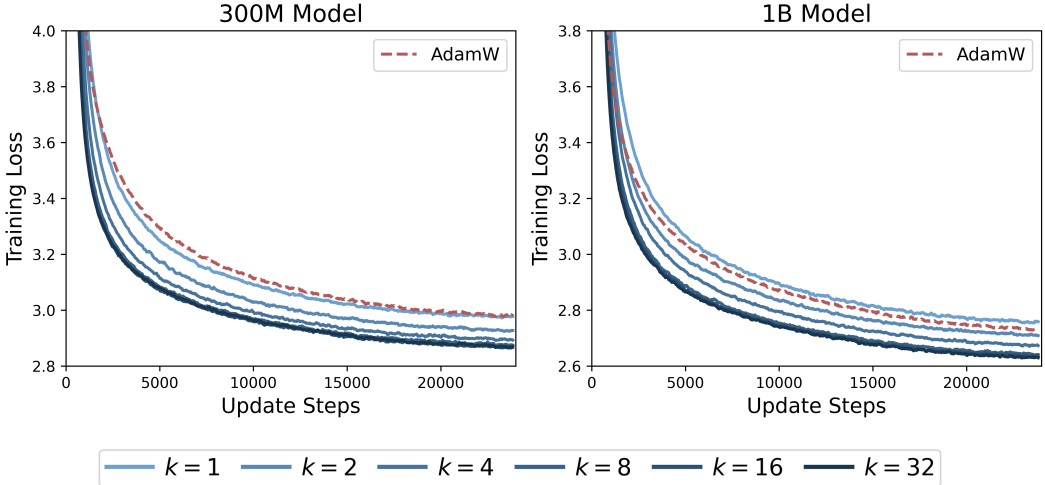

Figure 2: Training loss comparison between 1B and 300M parameter models across different top-k sparsity levels. The left panel shows the 300M model performance while the right panel shows the 1B model performance. Both models are trained with varying sparsification top $k$ (k=1, 2, 4, 8, 16, 32) compared against the AdamW baseline. The results demonstrate the effectiveness of top-k sparsification across different model sizes.

**Implementation.** DeMo integration required minimal changes to the codebase. We implemented a DeMo optimizer class and disabled default gradient synchronization in PyTorch Distributed Data Parallelism (Li et al., 2020). All experiments are conducted on 64 NVIDIA H100 GPUs with a global batch size of 2048 and a sequence length of 2048 tokens. We use four gradient accumulation steps, yielding an effective per-GPU batch size of 8.

## 3.2 PRETRAINING PERFORMANCE

**Convergence Behavior.** Figure 3.1 shows training loss curves comparing DeMo with various sparsification levels $k$ to the AdamW optimizer on both the OLMo-300M and OLMo-1B models. As the figure illustrates, a sparsification of just $k = 2$ is sufficient to achieve better training performance than AdamW. Increasing $k$ further provides marginal gains at a greater communication cost.

**Compute Efficiency.** DeMo provides a substantial reduction in communication. For instance, with the tensor chunk size of $64 \times 64$, selecting the top-$k$ values reduces upload bandwidth at each step by a factor of roughly $4096/k$. Figure 4 plots final model perplexity against the overall communication cost in MB per step.

| | **OLMo-300M** | | | | **OLMo-1B** | | | |
|---|---|---|---|---|---|---|---|---|
| **Optimizer / $k$** | Hella ↑ | ARC ↑ | PIQA ↑ | Tx ↓ | Hella ↑ | ARC ↑ | PIQA ↑ | Tx ↓ |
| | acc_n | acc | acc_n | MB/step | acc_n | acc | acc_n | MB/step |
| DeMo $k$=32 | 0.37 | 0.46 | 0.67 | 29.9 | 0.48 | 0.55 | 0.70 | 110.32 |
| DeMo $k$=16 | 0.38 | 0.50 | 0.67 | 14.9 | 0.47 | 0.53 | 0.70 | 55.16 |
| DeMo $k$=8 | 0.38 | 0.47 | 0.67 | 7.49 | 0.47 | 0.52 | 0.69 | 27.58 |
| DeMo $k$=4 | 0.37 | 0.47 | 0.67 | 3.74 | 0.45 | 0.52 | 0.70 | 13.79 |
| DeMo $k$=2 | 0.36 | 0.46 | 0.65 | 1.87 | 0.44 | 0.51 | 0.69 | 6.89 |
| DeMo $k$=1 | 0.35 | 0.45 | 0.65 | 0.93 | 0.41 | 0.52 | 0.69 | 3.44 |
| AdamW-DDP | 0.35 | 0.46 | 0.65 | 636.9 | 0.43 | 0.51 | 0.68 | 2416.6 |

Table 1: Zero-shot downstream accuracy and per-GPU communication volume (MB per training step) after 100B-token pretraining. Higher is better for accuracy metrics; lower is better for communication.

### 3.3 DOWNSTREAM EVALUATION

We assessed the pretrained checkpoints on three widely used zero-shot benchmarks: HellaSwag, ARC-Easy, and PIQA, and report normalized accuracy (or accuracy for ARC-Easy). Table 1 shows that DeMo matches or exceeds the AdamW-DDP baseline across all tasks while reducing per-GPU communication by two to three *orders of magnitude*.

For the 300M-parameter model, using $k = 8$ already cuts data transfer by 85 times (7.5 MB vs. 637 MB) with no loss in accuracy; smaller $k$ values offer additional savings with a minor accuracy trade-off. At the 1B scale, the trend persists: DeMo with $k = 16$ attains higher HellaSwag and PIQA scores than AdamW while transmitting only 55 MB per step—an efficiency gain of 44 times. These results confirm that our proposed decoupled momentum strategy preserves model quality even under aggressive communication budgets and scales favorably with model size.

### 3.4 ABLATION STUDIES

To better understand the impact of specific design choices in DeMo, we conducted comprehensive ablation studies:

**Impact of Momentum Subtraction.** An essential feature of DeMo is momentum subtraction. In this experiment, we systematically varied $\alpha$ from 0 (no subtraction) to 1 (full subtraction of communicated values). The results are shown in Figure 3.1. Clearly, no subtraction ($\alpha = 0$) is detrimental because, with a fixed basis, the top-$k$ elements evolve slowly and the same elements are repeatedly selected and communicated across steps, resulting in similar consecutive updates and degraded performance. Given that the momentum buffer also serves as an error accumulator, subtracting previously communicated values is necessary. However, compared to full subtraction, we found that using a smaller value ($\alpha = 0.2$) to gradually evolve the top-$k$ elements and partially decay communicated values over time further improves performance.

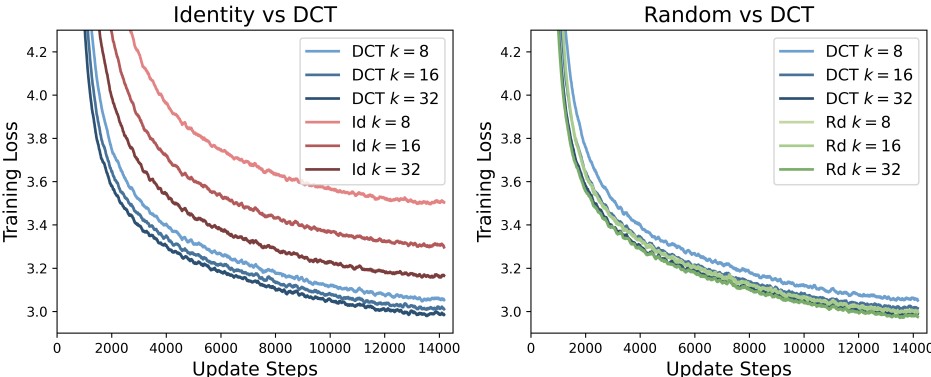

Figure 3: Training loss with different linear transformations. The left plot compares the DCT against the identity mapping, while the right plot compares the DCT with a random orthonormal projection matrices. Each curve corresponds to a different sparsity level $k \in \{8, 16, 32\}$.

**Choice of Linear Projections.** We explored three natural choices for the linear transformations $\mathcal{T}$. The baseline is the identity mapping $P_i = I$, i.e., no transformation. We compared this with the Discrete Cosine Transform (DCT)-based transformation, where $P_i$ corresponds to the DCT basis of dimension 64, and a random projection matrix, where each element of $P_i$ is sampled from $\mathcal{N}(0, I)$ and orthonormalized using the Gram-Schmidt process. We set the random seed equal to the step number on each worker, ensuring a unique random projection for every step but consistency across workers. Since all $P_i$ are orthonormal, we have $P_i^{-1} = P_i^{\top}$. We considered three sparsification levels: $k = 8, 16, 32$. The results are plotted in Figure 3.

As illustrated in the figure, the DCT-based transformation clearly improves performance over the identity mapping. This supports the hypothesis from related studies that naively updating parameters using sparse patterns can degrade performance, while applying a transformation enables pa-

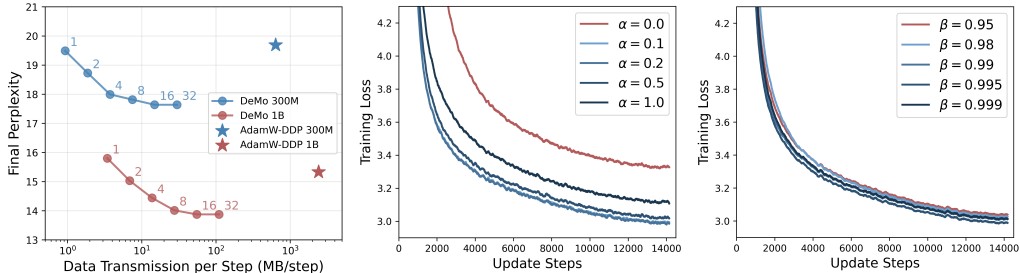

Figure 4: **Left:** Validation perplexity of OLMo-300M and OLMo-1B versus data transmitted per step (log scale; MB/step). Each DeMo point is annotated with the sparsification level $k$. **Middle:** Training loss of OLMo-300M with DeMo for momentum-subtraction magnitude $\alpha \in \{0.0, 0.1, 0.2, 0.5, 1.0\}$. **Right:** Training loss of DeMo with top-$k$=32, $C$=64, and $\alpha$=0.2 under varying momentum decay $\beta$.

rameter updates as linear combinations of many non-sparse vectors. Thereby the parameters are updated more uniformly, especially important when sparsity $k$ is low. Although random projection is arguably the most intuitive choice, as it continuously rotates and changes the momentum subspace perspective, we found that a fixed DCT basis yields comparable performance while eliminating the need to recompute the basis at each step.

### 3.5 ADDITIONAL EXPERIMENTS

This section presents additional ablations and comparisons of DeMo with communication-efficient optimizers, extensively tuned AdamW-DDP and Muon-DDP baselines, and momentum decay ablations. All experiments use a 300M parameter model trained with 20 tokens per parameter.

**Ablations on momentum coefficient $\beta$.** We ablate the momentum coefficient $\beta \in \{0.95, 0.98, 0.99, 0.995, 0.999\}$. Training curves are shown in Figure 4. Performance remains stable across this range, with larger values (0.995, 0.999) outperforming smaller ones. The best results are achieved at $\beta = 0.995$.

**Comparisons with AdamW-DDP and Muon-DDP.** To isolate optimizer effects, we reduce the token budget and perform extensive hyperparameter sweeps for AdamW-DDP. We search learning rates in $\{1.5 \times 10^{-3}, 1.2 \times 10^{-3}, 10^{-3}, 8 \times 10^{-4}, 6 \times 10^{-4}, 5 \times 10^{-4}, 3 \times 10^{-4}, 2 \times 10^{-4}\}$ and $\beta_2$ in $\{0.95, 0.98, 0.99, 0.995, 0.999\}$. For Muon-DDP, we use the recommended momentum value 0.95 in Liu et al. (2025) and tune the learning rate over the same grid. Figure 5 shows that Muon-DDP has better performance than AdamW-DDP. DeMo underperforms AdamW at equal update counts but provides substantially lower communication cost.

**Comparisons with DiLoCo (Douillard et al., 2023).** We additionally compare against DiLoCo, a communication-efficient method that alternates local updates with periodic global synchronization. Following the authors' recommended settings, we tune the outer-loop learning rate over $\{1.0, 0.7, 0.5, 0.3, 0.1\}$ for each run, and vary the global communication frequency to target different compression ratios. Figure 6 reports final validation perplexity versus data-receiving reduction when using eight workers. For example, for DeMo with a $64 \times 64$ chunk size and top-$k = 32$, the data-transmission and data-receiving reductions are 128 and 16, respectively (ignoring the marginal contribution of layer norms). At comparable compression levels, DeMo consistently yields lower perplexity than DiLoCo. Training-loss curves for DiLoCo with communication frequency 10 are included in Figure 5, where DeMo again shows superior optimization performance.

**Comparisons with PowerSGD (Vogels et al., 2019).** We also compare DeMo with PowerSGD, which reduces communication by applying a low rank projection during gradient aggregation. We use the official PyTorch implementation of the PowerSGD DDP communication hook (Paszke et al., 2019). Following the recommended procedure, we sweep the communication rank over the set $\{2, 4, 8, 16, 32, 64\}$ and observe that ranks below 16 lead to substantial degradation. We therefore

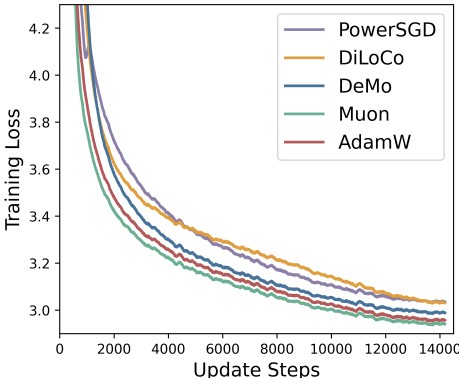
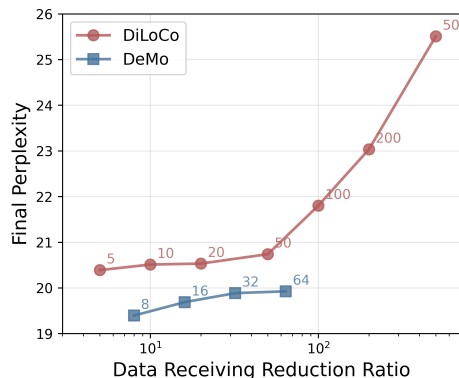

Figure 5: Training loss curves of four optimizers with tuned hyperparameters: AdamW, Muon, DeMo, DiLoCo and PowerSGD.

Figure 6: Data receiving compression ratio vs. final validation perplexity of DeMo and DiLoCo trained with varying top-$k$/communication interval.

select rank 32 as the preferred setting, which provides an effective compression ratio of approximately 26 times for the compressed steps. Warm up before compression is also important. As suggested in the implementation, we tune the number of warm up steps and find that using 1000 steps avoids the instabilities and divergence that often occur with only 500 steps. We search the learning rate over the same grid used for AdamW and find that the best value for PowerSGD is $6 \times 10^{-4}$, which is smaller than the optimal learning rate for AdamW. Because PowerSGD is essentially compressed SGD, which performs poorly for language model training in its basic form, we adopt an Adam style update rule using the compressed gradients. The final performance of PowerSGD is similar to DiLoCo but remains slightly below that of DeMo.

## 4 RELATED WORKS

Training large foundation models presents substantial communication bottlenecks, as synchronizing gradients across numerous accelerators can dominate training time (Shoeybi et al., 2019; Huang et al., 2019; Narayanan et al., 2021). To alleviate this, various gradient compression techniques have been developed (Aji and Heafield, 2017; Wen et al., 2017). Prominent among these are **sparsification** methods, which transmits only the gradients with the largest magnitudes (Lin et al., 2018b; Stich et al., 2018; Alistarh et al., 2018; Zhao et al., 2024). Another widely used approach is **gradient quantization**, which reduces the numerical precision of gradients (Alistarh et al., 2017; Sun et al., 2019; Zhao et al., 2024; Gorbunov et al., 2021; Peng et al., 2023). To compensate for information loss introduced by biased compressors, **error feedback** mechanisms, such as EF-SGD Karimireddy et al. (2019), locally accumulate the compression error and add it to the gradient in the subsequent iteration (Seide et al., 2014; Karimireddy et al., 2019). Our work also leverages top-k sparsification of momentum communication and leverages the error accumulation on momentum itself, thereby reducing one copy of memory for the accumulator. Recent work further explores *optimizer-aware* communication reductions (e.g., communication-efficient variants built around Lion-style updates) (Chen et al., 2024a; Chen, 2025; Chen et al., 2024b).

Another line of research aims to **reduce communication frequency** through decoupled optimization strategies. Methods like Federated Averaging (FedAvg) (McMahan et al., 2017) and Local SGD (Stich, 2018; Lin et al., 2018a) allow workers to perform multiple local updates before synchronizing model parameters, thereby reducing the number of communication rounds. However, these approaches can face challenges such as client drift, especially with non-iid data, which may impact convergence speed and final model quality (Karimireddy et al., 2020; Zhao et al., 2018). More recently, DiLoCo has emerged as an empirically successful technique for training large language models with significantly reduced communication (Douillard et al., 2023). While effective, the complex interplay of infrequent synchronization and local optimizer dynamics in such systems can sometimes lead to less predictable optimization trajectories (especially for modern adaptive/cautious variants) (Nguyen et al., 2025; Liang et al., 2024a; Chen et al., 2025b); our work instead

focuses on maintaining more frequent, albeit significantly compressed, synchronization of a critical optimizer state.

A third strategy seeks to reduce memory and communication via **low-rank updates**, moving beyond unstructured compression. This was popularized by Low-Rank Adaptation (LoRA) for parameter-efficient fine-tuning, which freezes pre-trained weights and learns a low-rank decomposition of the weight update matrix (Hu et al., 2022). More recently, this principle has been applied directly to the optimization process to enable training from scratch. For instance, GaLore and its variants project the full gradient onto a low-rank subspace before it is passed to the optimizer, thereby significantly reducing gradient memory overhead (Zhao et al., 2024; Hao et al., 2024). Related subspace/factorized optimization methods provide similar memory savings by restricting updates to a low-dimensional structure (Liang et al., 2024b; Nguyen et al., 2024). In distributed settings, related techniques communicate low-rank factors of the gradient or model update instead of the full matrix, directly reducing communication volume. Notable prior works include (Park and Klabjan, 2024; Zhao et al., 2025; Chen et al., 2025a). The top-k sparsification of DeMo operates in a fixed transformed space, thereby also making the communicated update matrix of each worker to be rank-k.

## 5 DISCUSSIONS AND LIMITATIONS

Our experimental findings confirm that DeMo consistently achieves competitive or improved convergence behavior compared to AdamW. Additionally, DeMo demonstrates reduced communication overhead and favorable downstream generalization. Our ablation studies further highlight the critical role of momentum sparsification with DCT, momentum subtraction and chunk-based projection techniques in balancing optimization performance and computational efficiency.

In our analysis, we primarily focused on the upload bandwidth per step. However, it is important to note that the download bandwidth scales with the number of workers. This limitation is not unique to our method but is intrinsic to all top-$k$ sparsification-based approaches, including those proposed in Lin et al. (2018b); Stich et al. (2018). DeMo is designed primarily for optimization across a small number of geographically distributed compute centers, enabling communication between optimization steps over the Internet rather than relying on specialized high-speed interconnects such as InfiniBand. This relaxes the need for dedicated long-distance networking infrastructure. Within each data center, standard DDP can still be employed; each center can be treated as a "large worker", and DeMo can then be used to efficiently coordinate communication between these centers.

## 6 CONCLUSION

In conclusion, we have shown that our proposed DeMo optimization algorithm can act as a drop-in replacement to AdamW when training LLMs, with no noticeable slowdown in convergence while reducing communication requirements by several orders of magnitude. The signum variant of DeMo is more memory efficient than AdamW and has negligible compute overhead if we use small precomputed DCT transition matrices. Finally, the LLMs pre-trained with DeMo have equivalent or better scores on multiple standard benchmarks compared to their equivalents trained with AdamW.

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

# 7 USE OF LARGE LANGUAGE MODELS

Regarding paper writing, we used LLM only for text polishing and grammar correction during manuscript preparation. No LLMs were involved in the conception or design of the method, experiments, or analysis. All technical content, results, and conclusions have been independently verified and validated by the authors.

# 8 CONVERGENCE ANALYSIS

In this section, we establish the convergence guarantees of the DeMo algorithm under standard assumptions: unbiased stochastic gradients with bounded variance (Assumption 4), $L$-smoothness of the objective (Assumption 5), and bounded gradient norms (Assumption 6). The main result is Theorem 2, which proves an $O(1/\sqrt{T})$ convergence rate in terms of the average gradient norm. We further quantify the bias and approximation error introduced by sparse top-$k$ averaging (Lemmas 2 and 8.1), which are crucial in analyzing the impact of sparsified communication.

Starting from SGD with Momentum, we make two key modifications: first, we remove the all-reduce operation on gradients $\tilde{g}_k$, decoupling momentum $m$ across the accelerators. Second, after updating the momentum, we extract and remove its fast components $q$, which can be efficiently synchronized with minimal communication. Algorithm 4 presents the complete method:

---

**Algorithm 2** Decoupled Momentum Optimization

---

**Input:** learning rate $\eta$, decay $\beta \in (0, 1)$, parameters $x_t$, momentum $m_t$, hyperparameters $s, k$
$\tilde{g}_t \leftarrow \text{LocalStochasticGradient}(x_t)$ {Get Local Gradient $g$ Without All-Reduce}
$m_t \leftarrow \beta m_t + \tilde{g}_t$ {Accumulate Gradient in Momentum $m$}
$q_t \leftarrow \text{ExtractFastComponents}(m_t, s, k)$ {Extract Fast Components $q$ From $m$}
$m_{t+1} \leftarrow m_t - q_t$ {Remove $q$ From $m$}
$Q_t \leftarrow \text{Synchronize}(q_t)$ {Synchronize $q$ Across All Accelerators}
$x_{t+1} \leftarrow x_t - \eta Q_t$ {Parameter Update Step}

---

## 8.1 PRELIMINARIES

### 8.1.1 NOTATIONS

We summarize the key notation used throughout the convergence analysis in Table 2.

### 8.1.2 MULTILINEAR TRANSFORMS AND CHUNK-WISE TENSOR BLOCKING

We begin by introducing the tensorial operations central to our algorithmic design. Specifically, we define a multilinear product $\mathcal{T}$ acting on tensors via separable linear transforms, and describe the blocking operator $\mathcal{B}$ used to partition tensors into contiguous chunks. These constructions allow us to formalize the application of the Discrete Cosine Transform (DCT) and its inverse in a chunk-wise manner, which underpins the sparsification mechanism employed in the DeMo algorithm. We also introduce the key intermediate tensors used in the algorithm's updates, along with relevant dimensional and notational conventions.

Let $\mathcal{X} \in \mathbb{R}^{s_0 \times s_1 \times \cdots \times s_{d-1}}$ be a tensor of order $d$, and let matrices $P_i \in \mathbb{R}^{s_i \times s_i}$ be given for each $i \in [d]_0 = \{0, 1, \ldots, d-1\}$. We define the multilinear product $\mathcal{T}$ as follows:

$$\mathcal{T}(\mathcal{X}; P_0, P_1, \ldots, P_{d-1}) \in \mathbb{R}^{s_0 \times s_1 \times \cdots \times s_{d-1}},$$

whose entries are explicitly given by

$$\mathcal{T}(\mathcal{X}; P_0, \ldots, P_{d-1})_{i_0 i_1 \ldots i_{d-1}} = \sum_{j_0=1}^{s_0} \sum_{j_1=1}^{s_1} \cdots \sum_{j_{d-1}=1}^{s_{d-1}} \left( \prod_{k=0}^{d-1} (P_k)_{i_k j_k} \right) \mathcal{X}_{j_0 j_1 \ldots j_{d-1}}.$$

In the special case $d = 2$, this definition reduces to the familiar matrix multiplication form:

$$\mathcal{T}(\mathcal{X}; P_0, P_1) = P_0 \mathcal{X} P_1^\top.$$

Table 2: Summary of notation used in the convergence analysis.

| Notation | Definition |
|---|---|
| $\mathcal{X}, \mathbf{X}$ | Tensor in $\mathbb{R}^{n_0 \times \cdots \times n_{d-1}}$ |
| $d$ | Tensor order (number of dimensions) |
| $n_i$ | Size of tensor along dimension $i$ |
| $s_i$ | Chunk size along dimension $i$ |
| $c_i$ | Number of chunks along dimension $i$, $n_i = c_i s_i$ |
| $P_i$ | Transformation matrix (e.g., DCT) for dimension $i$, $P_i \in \mathbb{R}^{s_i \times s_i}$ |
| $\mathcal{T}$ | Multilinear tensor product operator |
| $\mathcal{B}$ | Blocking operator (tensor partition into chunks) |
| $\mathbf{k}$ | Chunk index: $\mathbf{k} = (k_0, \ldots, k_{d-1})$, with $k_i \in [c_i]_0$ |
| $B_{\mathbf{k}}(\mathcal{X})$ | Chunk indexed by $\mathbf{k}$: $B_{\mathbf{k}}(\mathcal{X}) = \mathcal{X}[k_0 s_0 : (k_0 + 1)s_0, \ldots, k_{d-1}s_{d-1} : (k_{d-1} + 1)s_{d-1}]$ |
| $\mathbf{M}_t^i$ | Local momentum at worker $i$, iteration $t$ |
| $\mathbf{M}_t$ | Global momentum |
| $\mathbf{G}_t^i$ | Stochastic gradient at worker $i$, iteration $t$ |
| $\mathbf{P}_t^i$ | Intermediate tensor: $\mathbf{P}_t^i = \beta \mathbf{M}_{t-1}^i + \mathbf{G}_t^i$ |
| $\hat{\mathbf{P}}_t^i$ | Chunk-wise DCT of intermediate tensor $\mathbf{P}_t^i$ |
| $\mathbf{Q}_t^i$: | Perform chunk-wise top-$k$ selection on $\hat{\mathbf{P}}_t^i$, followed by chunk-wise inverse DCT |
| $\mathbf{P}_t$ | Aggregated intermediate tensor: $\mathbf{P}_t = \beta \mathbf{e}_t + \frac{1}{N} \sum_{i=1}^{N} \left( \mathbf{Q}_t^i + (1 - \beta)\mathbf{M}_{t+1}^i \right)$ |
| $\Phi_t$ | Aggregated tensor after sparse top-$k$ aggregation, iteration $t$ |
| $\mathbf{X}_t$ | Model Parameter at iteration $t$ |
| $\beta$ | Momentum decay parameter ($0 < \beta < 1$) |
| $\eta$ | Learning rate |
| $\lambda$ | Weight decay parameter |
| $k$ | Sparsity parameter (number of elements retained per chunk) |
| $D$ | Total number of tensor elements: $D = \prod_{j=0}^{d-1} n_i$ |
| $M$ | Number of elements per chunk: $M = \prod_{i=0}^{d-1} s_i$ |
| $\mathcal{L}(\mathbf{X})$ | Objective function |
| $\mathcal{L}^*$ | Optimal objective value |
| $\xi, \xi_i$ | Random samples from data distribution $\mathcal{D}$ or $\mathcal{D}_i$ |
| $\sigma^2$ | Variance bound of stochastic gradients |
| $N$ | Number of distributed workers |
| $\mathcal{D}_i$ | Dataset at worker $i$ |

It is evident that the Discrete Cosine Transform (DCT) represents a specific instance of the multi-linear product, alongside its counterpoint, the Inverse Discrete Cosine Transform (IDCT). Both are defined as unitary matrices, leading to the equality $P_i^\top = P_i^{-1}$ for each $i \in [d]_0$.

Consider now a tensor $\mathcal{X} \in \mathbb{R}^{n_0 \times n_1 \times \cdots \times n_{d-1}}$. Suppose each dimension $n_i$ is divisible by $s_i$, so we can write $n_i = c_i s_i$ for all $i \in [d]_0$. We define the *blocking operator* $\mathcal{B}$ acting on $\mathcal{X}$ as:

$$\mathcal{B}(\mathcal{X}) = \{B_{\mathbf{k}}(\mathcal{X})\}_{\mathbf{k} \in [c_0]_0 \times [c_1]_0 \times \cdots \times [c_{d-1}]_0},$$

where each *block* $B_{\mathbf{k}}(\mathcal{X}) \in \mathbb{R}^{s_0 \times s_1 \times \cdots \times s_{d-1}}$ is explicitly defined elementwise by:

$$(B_{\mathbf{k}}(\mathcal{X}))_{\mathbf{j}} = \mathcal{X}_{k_0 s_0 + j_0,\ k_1 s_1 + j_1,\ \ldots,\ k_{d-1}s_{d-1} + j_{d-1}},$$

with indices:

- $\mathbf{k} = (k_0, k_1, \ldots, k_{d-1})$ specifying the position of the block, where $k_i \in [c_i]_0$.
- $\mathbf{j} = (j_0, j_1, \ldots, j_{d-1})$ indexing within each block, with $j_i \in [s_i]_0$.

**Example (Matrix Blocking).** Consider a matrix $\mathcal{X} \in \mathbb{R}^{128 \times 512}$. We partition $\mathcal{X}$ into contiguous submatrices (blocks) of size $16 \times 32$. Thus, each dimension is factorized as:

$$n_0 = 128 = 8 \times 16, \quad n_1 = 512 = 16 \times 32.$$

We define the blocking operator $\mathcal{B}$ applied to $\mathcal{X}$ as:

$$\mathcal{B}(\mathcal{X}) = \{B_{k_0, k_1}(\mathcal{X})\}_{k_0 \in [8]_0,\ k_1 \in [16]_0},$$

where each block $B_{k_0,k_1}(\mathcal{X}) \in \mathbb{R}^{16 \times 32}$ is explicitly given by:

$$(B_{k_0,k_1}(\mathcal{X}))_{j_0,j_1} = \mathcal{X}_{k_0 \cdot 16 + j_0, \, k_1 \cdot 32 + j_1},$$

with indices $k_0 \in \{0, 1, \ldots, 7\}$, $k_1 \in \{0, 1, \ldots, 15\}$ denoting the block position, and local indices within each block given by $j_0 \in \{0, 1, \ldots, 15\}$, $j_1 \in \{0, 1, \ldots, 31\}$.

For instance, the upper-leftmost block $B_{0,0}(\mathcal{X})$ has entries:

$$B_{0,0}(\mathcal{X}) = \begin{pmatrix} \mathcal{X}_{0,0} & \mathcal{X}_{0,1} & \ldots & \mathcal{X}_{0,31} \\ \mathcal{X}_{1,0} & \mathcal{X}_{1,1} & \ldots & \mathcal{X}_{1,31} \\ \vdots & \vdots & \ddots & \vdots \\ \mathcal{X}_{15,0} & \mathcal{X}_{15,1} & \ldots & \mathcal{X}_{15,31} \end{pmatrix}.$$

Let $\mathbf{X}$ be a $d$-dimensional tensor, partitioned into chunks of shape $(s_0 \times \cdots \times s_{d-1})$. We denote an individual chunk of $\mathbf{X}$ by $B_{\mathbf{k}}(\mathbf{X})$, where $\mathbf{k}$ is a multi-index ranging over the set of all chunk indices $\mathbf{K}$.

We define an intermediate tensor $\mathbf{P}_t^i$ as

$$\mathbf{P}_t^i := \beta \mathbf{M}_{t-1}^i + \mathbf{G}_t^i.$$

Next, we apply a chunk-wise Discrete Cosine Transform (DCT), denoted by $\mathcal{T}$, to each chunk of $\mathbf{P}_t^i$, yielding the transformed tensor $\hat{\mathbf{P}}_t^i$. This operation is defined as

$$B_{\mathbf{k}}(\hat{\mathbf{P}}_t^i) := \mathcal{T}\left(B_{\mathbf{k}}(\mathbf{P}_t^i)\right) \quad \text{for all } \mathbf{k} \in \mathbf{K}.$$

Finally, for notational convenience, we define the total number of elements in the original and chunk shapes as

$$D = \prod_{i=0}^{d-1} n_i, \quad M = \prod_{i=0}^{d-1} s_i.$$

By default, we let $\|\cdot\|$ denote the $\ell_2$ norm unless otherwise specified. We denote $\Phi_t$ as the aggregated tensor obtained via a scatter all-reduce operation over the encoded tensors $\mathbf{Q}_t^i$ from each worker as detailed in 3.

### 8.1.3 PROBLEM SETTINGS

In general, we consider minimizing the following objective function:

$$\min_{\mathbf{X} \in \mathbb{X}} \mathcal{L}(\mathbf{X}) := \mathbb{E}_{\xi \sim \mathcal{D}} \left[\mathcal{L}(\mathbf{X}, \xi)\right], \tag{9}$$

where $\mathbb{X} = \mathbb{R}^{n_0 \times n_1 \times \cdots \times n_{d-1}}$, and $\mathcal{L}(\mathbf{X}, \xi)$ is a general loss function, and the expectation is taken over the data distribution $\mathcal{D}$ from which samples $\xi$ are drawn. We denote by $\mathcal{L}^* := \inf_{\mathbf{X} \in \mathbb{X}} \mathcal{L}(\mathbf{X})$ the optimal value of the objective function and assume throughout this paper that $\mathcal{L}^*$ is finite and bounded from below. Given a realization $\mathcal{L}(\mathbf{X}, \xi)$, the *stochastic gradient* $\nabla \mathcal{L}(\mathbf{X}, \xi)$ is defined as the gradient of $\mathcal{L}(\mathbf{X}, \xi)$ with respect to the parameter vector $\mathbf{X}$.

In the distributed training setting, we aim to solve the optimization problem:

$$\min_{\mathbf{X} \in \mathbb{X}} \mathcal{L}(\mathbf{X}) := \frac{1}{N} \sum_{i=1}^{N} \mathbb{E}_{\xi_i \sim \mathcal{D}_i} \left[\mathcal{L}(\mathbf{X}, \xi_i)\right], \tag{10}$$

where $N$ denotes the number of workers, and $\{\mathcal{D}_i\}_{i=1}^N$ represent the datasets available at each worker.[2] Here, $\mathbf{X}$ represents the model parameters (e.g., neural network weights). Under this distributed scenario, each worker $i \in [N]$ maintains its own dataset $\mathcal{D}_i$, and there is a centralized server accessible to all workers for communication. At training step $t$, the stochastic gradient $\mathbf{G}_t^i := \nabla \mathcal{L}(\mathbf{X}_t, \xi_t^i)$ at worker $i$ is computed using a data batch $\xi_t^i$ sampled from the dataset $\mathcal{D}_i$.

---

[2]Throughout this work, we assume datasets $\{\mathcal{D}_i\}_{i=1}^N$ consist of i.i.d. samples, and each sample $\xi_i \sim \mathcal{D}_i$ is drawn independently. However, our proposed method can directly extend to non-i.i.d. settings.

**Definition 1** (Variance). *The* variance *of a $\mathbb{X}$-valued random variable $\mathbf{X}$ is defined as*

$$\text{Var}(\mathbf{X}) := \mathbb{E}\left[\|\mathbf{X} - \mathbb{E}[\mathbf{X}]\|_{\text{F}}^2\right].$$

With the definition of variance, we have the following assumption used for the analysis of stochastic settings.

**Assumption 4.** *The stochastic samples $\xi \sim \mathcal{D}$ are independent and identically distributed (i.i.d.). Additionally, the stochastic gradient $\nabla\mathcal{L}(\mathbf{X}, \xi)$ satisfies:*

$$\mathbb{E}[\nabla\mathcal{L}(\mathbf{X}, \xi)] = \nabla\mathcal{L}(\mathbf{X}), \quad and \quad \text{Var}(\nabla\mathcal{L}(\mathbf{X}, \xi)) \leq \frac{\sigma^2}{n_{\text{batch}}},$$

*where $\sigma^2$ is a finite constant and $n_{\text{batch}}$ denotes the batch size.*

Assumption 4 ensures that the variance of the stochastic gradient is uniformly bounded. For our discrete-time analysis, we also utilize the $L$-smoothness condition stated in Assumption 5.

**Assumption 5** ($L$-smoothness). *The objective function $\mathcal{L}(\mathbf{X})$ is differentiable, lower-bounded (i.e., $\mathcal{L}^* = \inf_{\mathbf{X} \in \mathbb{X}} \mathcal{L}(\mathbf{X}) > -\infty$), and $L$-smooth.*

We say that a differentiable function $\mathcal{L} : \mathbb{X} \to \mathbb{R}$ is *L-smooth* if for all $\mathbf{X}, \mathbf{Y} \in \mathbb{X}$,

$$\|\nabla\mathcal{L}(\mathbf{Y}) - \nabla\mathcal{L}(\mathbf{X})\|_{\text{F}} \leq L \|\mathbf{Y} - \mathbf{X}\|_{\text{F}}.$$

If $\mathcal{L}$ is $L$-smooth, then for all $\mathbf{X}, \mathbf{Y} \in \mathbb{X}$, we have

$$\mathcal{L}(\mathbf{Y}) \leq \mathcal{L}(\mathbf{X}) + \langle\nabla\mathcal{L}(\mathbf{X}), \mathbf{Y} - \mathbf{X}\rangle + \frac{L}{2}\|\mathbf{Y} - \mathbf{X}\|_{\text{F}}^2.$$

**Assumption 6** (Bounded Gradient). *For any $\mathbf{X} \in \mathbb{X}$, and $\xi \sim \mathcal{D}$, the stochastic gradient satisfies $\mathbb{E}[\|\nabla f(\mathbf{X}; \xi)\|_1] \leq R$ with $R > 0$.*

Assumptions 4, 5 and 6 are standard in the analysis of stochastic optimization algorithms Bernstein and Newhouse (2024); Défossez et al. (2022); Liu et al. (2024b).

We employ a sparse aggregation protocol as a form of all-reduce. The procedure for calculating the final aggregated vector is as follows:

---
**Algorithm 3** Sparse Top-$k$ Averaging

---
1: **Input:** A set of tensors $\{X^{(j)}\}_{j=1}^N$, where $X^{(j)} \in \mathbb{R}^d$; sparsity parameter $k \in \{1, \ldots, d\}$.
2: **Output:** An aggregated tensor $\bar{X} \in \mathbb{R}^d$.
3: Initialize sum $S \in \mathbb{R}^d$ and counts $n \in \mathbb{Z}^d$ to zeros.
4: **for** $j = 1, \ldots, N$ **do**
5:    $\mathcal{I}_j \leftarrow \text{TopKIndices}(|X^{(j)}|, k)$
6:    $S_{\mathcal{I}_j} \leftarrow S_{\mathcal{I}_j} + X_{\mathcal{I}_j}^{(j)}$
7:    $n_{\mathcal{I}_j} \leftarrow n_{\mathcal{I}_j} + 1$
8: **end for**
9: $\bar{X} \leftarrow S \oslash n$, where $\oslash$ denotes element-wise division with $0/0 = 0$.
10: **return** $\bar{X}$

---

With the problem settings and mitigating strategies discussed above, we now proceed to the analysis of convergence and communication complexity of our proposed Decoupled Momentum Optimization (DeMO) algorithm.

## 8.2 CONVERGENCE ANALYSIS

In this section, we provide the convergence analysis for the proposed Decoupled Momentum Optimization (DeMO) algorithm. We make use of the standard smoothness, variance and bounded gradient assumptions, as stated in Assumptions 4,5, and 6, in the analysis.

---

**Algorithm 4** Decoupled Momentum Optimization

---

1: **Given:** learning rate $\eta$, momentum decay rate $\beta$, weight decay $\lambda$, chunk sizes $(s_0, \ldots, s_{d-1})$, sparsity $k$, DCT matrices $\{P_i\}_{i=0}^{d-1}$
2: **Initialize:** $t \leftarrow 0$, parameters $\mathbf{X}_0$, global momentum $\mathbf{M}_0$
3: **repeat**
4:     $t \leftarrow t + 1$
5:     **Worker Side (in parallel for each worker $i$):**
6:         Maintain local momentum residual $\mathbf{M}_{t-1}^i$
7:         Compute gradient $\mathbf{G}_t^i \leftarrow \nabla\mathcal{L}(\mathbf{X}_{t-1}, \xi_{t-1}^i)$
8:         Update local momentum: $\mathbf{M}_t^i \leftarrow \beta\mathbf{M}_{t-1}^i + \mathbf{G}_t^i$
9:     **for** each sub-chunk $B_\mathbf{k}(\mathbf{M}_t^i)$ of $\mathbf{M}_t^i$ **do**
10:         Compute DCT: $\mathbf{Q}_{\mathbf{k},t}^i \leftarrow \mathcal{T}(B_\mathbf{k}(\mathbf{M}_t^i), P_0, \ldots, P_{d-1})$
11:         Threshold thr $\leftarrow k$-th largest magnitude in $|\mathbf{Q}_{\mathbf{k},t}^i|$
12:         Sparsify: MASK $\leftarrow |\mathbf{Q}_{\mathbf{k},t}^i| \geq$ thr
13:         $\mathbf{Q}_{\mathbf{k},t}^i \leftarrow \mathbf{Q}_{\mathbf{k},t}^i \odot$ MASK
14:         Update residual: $B_\mathbf{k}(\mathbf{M}_t^i) \leftarrow B_\mathbf{k}(\mathbf{M}_t^i) - \mathcal{T}(\mathbf{Q}_{\mathbf{k},t}^i, P_0^\top, \ldots, P_{d-1}^\top)$
15:     **end for**
16:     Send sparse encode($\mathbf{Q}_{\mathbf{k},t}^i$) to server
17:     **Server Side:**
18:     **for** each sub-chunk $B_\mathbf{k}(\mathbf{M}_t)$ of global momentum $\mathbf{M}_t$ **do**
19:         Aggregate: $B_\mathbf{k}(\mathbf{M}_t) \leftarrow$ Aggregate$\{$encode($\mathbf{Q}_{\mathbf{k},t}^i$)$\}_i$
20:         Inverse DCT: $B_\mathbf{k}(\mathbf{M}_t) \leftarrow \mathcal{T}(B_\mathbf{k}(\mathbf{M}_t), P_0^\top, \ldots, P_{d-1}^\top)$
21:     **end for**
22:     Update parameters: $\mathbf{X}_t \leftarrow \mathbf{X}_{t-1} - \eta(\text{sgn}(\mathbf{M}_t) + \lambda\mathbf{X}_{t-1})$
23:     Broadcast sgn($\mathbf{M}_t$) to workers
24:     **Worker Side:**
25:         Receive and update local parameters with sgn($\mathbf{M}_t$)
26: **until** stopping criterion met
27: **return** optimized parameters $\mathbf{X}_t$

---

**Theorem 2** (Convergence of DeMo). *Under Assumptions 6, 4, and 5, let $\{\mathbf{X}_t\}_{t=1}^T$ be the sequence generated by Algorithm 4. Then, the following convergence bound holds:*

$$\frac{1}{T}\sum_{t=1}^T \mathbb{E}\left[\|\nabla\mathcal{L}(\mathbf{X}_t)\|_1\right] \leq \frac{\mathbb{E}[\mathcal{L}(\mathbf{X}_0) - \mathcal{L}(\mathbf{X}_T)]}{T\eta} + 2LD\eta + \frac{\sigma}{\sqrt{Nn_{\text{batch}}}}$$

$$+ R\sqrt{D}\frac{\sqrt{1 - \frac{k}{M}}}{1 - \beta\sqrt{1 - \frac{k}{M}}}\left(\beta + 2D\sqrt{\frac{2\pi}{N}}\left(1 + \sqrt{\frac{M}{k}}\right)\right).$$

**Remark 1** (Special Cases and Convergence Rates). *(i) In the special case where $k = M$, the last term vanishes, simplifying the bound to*

$$\frac{1}{T}\sum_{t=1}^T \mathbb{E}\left[\|\nabla\mathcal{L}(\mathbf{X}_t)\|_1\right] \leq \frac{\mathbb{E}[\mathcal{L}(\mathbf{X}_0) - \mathcal{L}(\mathbf{X}_T)]}{T\eta} + 2LD\eta + \frac{\sigma}{\sqrt{Nn_{\text{batch}}}}.$$

*Choosing the step size $\eta = \Theta\left(\frac{1}{\sqrt{T}}\right)$, we achieve the convergence rate:*

$$\frac{1}{T}\sum_{t=1}^T \mathbb{E}\left[\|\nabla\mathcal{L}(\mathbf{X}_t)\|_1\right] = O\left(\frac{1}{\sqrt{T}}\right) + O\left(\frac{1}{\sqrt{N}}\right).$$

*(ii) Alternatively, if we choose $\beta = \sqrt{1/T}$, the bound becomes*

$$\frac{1}{T}\sum_{t=1}^{T}\mathbb{E}\left[\|\nabla\mathcal{L}(\mathbf{X}_t)\|_1\right] \leq \frac{\mathbb{E}[\mathcal{L}(\mathbf{X}_0) - \mathcal{L}(\mathbf{X}_T)]}{T\eta} + 2LD\eta + \frac{\sigma}{\sqrt{Nn_{\text{batch}}}}$$

$$+ R\sqrt{D}\frac{\sqrt{1 - \frac{k}{M}}}{1 - \sqrt{\frac{1}{T}\left(1 - \frac{k}{M}\right)}}\left(\sqrt{\frac{1}{T}} + 2D\sqrt{\frac{2\pi}{N}}\left(1 + \sqrt{\frac{M}{k}}\right)\right).$$

*With the additional choice of step size $\eta = \Theta\left(\frac{1}{\sqrt{T}}\right)$, the convergence rate is again:*

$$\frac{1}{T}\sum_{t=1}^{T}\mathbb{E}\left[\|\nabla\mathcal{L}(\mathbf{X}_t)\|_1\right] = O\left(\frac{1}{\sqrt{T}}\right) + O\left(\frac{1}{\sqrt{N}}\right).$$

*Proof.* Let us first decompose the difference in the loss function $\mathcal{L}$ using the $L$-smoothness property:

$$\mathcal{L}(\mathbf{X}_{t+1}) - \mathcal{L}(\mathbf{X}_t) \leq \langle\nabla\mathcal{L}(\mathbf{X}_t), \mathbf{X}_{t+1} - \mathbf{X}_t\rangle + \frac{L}{2}\|\mathbf{X}_{t+1} - \mathbf{X}_t\|_F^2 \quad (\textit{L-smoothness of } \mathcal{L})$$

$$= -\eta\langle\nabla\mathcal{L}(\mathbf{X}_t), \text{sign}(\mathbf{M}_{t+1})\rangle + \frac{L}{2}\|\mathbf{X}_{t+1} - \mathbf{X}_t\|_F^2$$

$$= -\eta\langle\nabla\mathcal{L}(\mathbf{X}_t), \text{sign}(\nabla\mathcal{L}(\mathbf{X}_t))\rangle + \frac{L}{2}\|\mathbf{X}_{t+1} - \mathbf{X}_t\|_F^2 \qquad (11)$$

$$+ \eta\langle\nabla\mathcal{L}(\mathbf{X}_t), \text{sign}(\nabla\mathcal{L}(\mathbf{X}_t)) - \text{sign}(\mathbf{M}_{t+1})\rangle$$

$$\leq -\eta\|\nabla\mathcal{L}(\mathbf{X}_t)\|_1 + 2L\eta^2 d$$

$$+ \eta\langle\nabla\mathcal{L}(\mathbf{X}_t), \text{sign}(\nabla\mathcal{L}(\mathbf{X}_t)) - \text{sign}(\mathbf{M}_{t+1})\rangle,$$

where we used the inequality

$$\|\mathbf{X}_{t+1} - \mathbf{X}_t\|_F^2 = \eta^2\|\text{sign}(\mathbf{M}_{t+1})\|_F^2 \leq \eta^2 d.$$

Using Lemma 8.3 with $a = 1$, we bound the last term as

$$\mathbb{E}\left[\langle\nabla\mathcal{L}(\mathbf{X}_t), \text{sign}(\nabla\mathcal{L}(\mathbf{X}_t)) - \text{sign}(\mathbf{M}_{t+1})\rangle\right] \leq 2\,\mathbb{E}\|\nabla\mathcal{L}(\mathbf{X}_t) - \mathbf{M}_{t+1}\|_1. \qquad (12)$$

To bound the term $\mathbb{E}\|\nabla\mathcal{L}(\mathbf{X}_t) - \mathbf{M}_{t+1}\|_1$, we apply the triangle inequality:

$$\|\nabla\mathcal{L}(\mathbf{X}_t) - \mathbf{M}_{t+1}\|_1 = \left\|\underbrace{\nabla\mathcal{L}(\mathbf{X}_t) - \frac{1}{N}\sum_{i=1}^{N}(\beta\mathbf{M}_t^i + \nabla\mathcal{L}(\mathbf{X}_t, \xi_t^i))}_{T_1} + \underbrace{\frac{1}{N}\sum_{i=1}^{N}(\beta\mathbf{M}_t^i + \nabla\mathcal{L}(\mathbf{X}_t, \xi_t^i)) - \mathbf{M}_{t+1}}_{T_2}\right\|_1$$

$$\leq \|T_1\|_1 + \|T_2\|_1.$$

We separately bound the terms $T_1$ and $T_2$.

**Bounding $T_1$:** Notice that

$$T_1 = \nabla\mathcal{L}(\mathbf{X}_t) - \underbrace{\frac{1}{N}\sum_{i=1}^{N}\nabla\mathcal{L}(\mathbf{X}_t, \xi_t^i)}_{(*)} - \underbrace{\frac{1}{N}\sum_{i=1}^{N}\beta\mathbf{M}_t^i}_{(**)}.$$

To bound the first term $(*)$, we recall:

$$(*) = \nabla\mathcal{L}(\mathbf{X}_t) - \frac{1}{N}\sum_{i=1}^{N}\nabla\mathcal{L}(\mathbf{X}_t, \xi_t^i).$$

By Assumption 4, the stochastic gradients $\nabla\mathcal{L}(\mathbf{X}_t, \xi_t^i)$ have bounded variance, i.e.,

$$\mathbb{E}\left[\|\nabla\mathcal{L}(\mathbf{X}_t, \xi_t^i) - \nabla\mathcal{L}(\mathbf{X}_t)\|_2^2\right] \leq \frac{\sigma^2}{n_{\text{batch}}}.$$

Thus, applying Jensen's inequality (from $\ell_2$ to $\ell_1$ norms) and independence across $i$, we have:

$$\mathbb{E}[\|(*)\|_1] \leq \sqrt{D}\,\mathbb{E}[\|(*)\|_2] \leq \sqrt{D}\sqrt{\mathbb{E}[\|(*)\|_2^2]} = \sqrt{D}\sqrt{\frac{1}{N}\frac{\sigma^2}{n_{\text{batch}}}} = \frac{\sigma\sqrt{D}}{\sqrt{N\,n_{\text{batch}}}}.$$

For the second term $(**)$, we have

$$(**) = \frac{1}{N}\sum_{i=1}^{N}\beta\mathbf{M}_t^i.$$

Since the tensors $\{\mathbf{M}_t^i\}_{i=1}^N$ are identically distributed, it follows that

$$\mathbb{E}[\|(**)\|_1] \leq \frac{1}{N}\sum_{i=1}^{N}\beta\,\mathbb{E}[\|\mathbf{M}_t^i\|_1] = \beta\,\mathbb{E}[\|\mathbf{M}_t^i\|_1].$$

Applying Lemma 1, we can further bound this expectation as:

$$\mathbb{E}[\|(**)\|_1] \leq \beta\sqrt{D}\,\mathbb{E}[\|\mathbf{M}_t^i\|_2] \leq \frac{\beta R\sqrt{D(1-k/M)}}{1 - \beta\sqrt{1-k/M}}.$$

**Bounding $T_2$:** Using Lemma 2, we have

$$\mathbb{E}[\|T_2\|_1] \leq \sqrt{D}\,\mathbb{E}[\|T_2\|_2] = \sqrt{D}\,\mathbb{E}\left[\|\Phi_{t+1} - \frac{1}{N}\sum_{i=1}^{N}\hat{\mathbf{P}}_t^i\|_2\right] \quad \text{(since DCT is orthonormal)}$$

$$\leq \sqrt{D}\sum_{\mathbf{k}\in\mathbf{K}}\mathbb{E}\left[\|B_{\mathbf{k}}(\Phi_{t+1}) - \frac{1}{N}\sum_{i=1}^{N}B_{\mathbf{k}}(\hat{\mathbf{P}}_t^i)\|_1\right]$$

$$\leq 2D^{3/2}R\sqrt{\frac{2\pi}{N}}\left(1 + \sqrt{\frac{M}{k}}\right)\frac{\sqrt{1-\frac{k}{M}}}{1 - \beta\sqrt{1-\frac{k}{M}}}, \quad \text{(by Lemma 2)}$$

Combining these results, we conclude

$$\mathbb{E}[\|\nabla\mathcal{L}(\mathbf{X}_t) - \mathbf{M}_{t+1}\|_1] \leq \frac{\sigma\sqrt{D}}{\sqrt{N n_{\text{batch}}}} + R\sqrt{D}\frac{\sqrt{1-k/M}}{1 - \beta\sqrt{1-k/M}}\left(\beta + 2D\sqrt{\frac{2\pi}{N}}\left(1 + \sqrt{\frac{M}{k}}\right)\right).$$

Hence, from equation 11, we obtain

$$\frac{1}{T}\sum_{t=1}^{T}\mathbb{E}[\|\nabla\mathcal{L}(\mathbf{X}_t)\|_1] \leq \frac{\mathbb{E}[\mathcal{L}(\mathbf{X}_0) - \mathcal{L}(\mathbf{X}_T)]}{T\eta} + 2LD\eta + \frac{\sigma}{\sqrt{N n_{\text{batch}}}}$$

$$+ R\sqrt{D}\frac{\sqrt{1-k/M}}{1 - \beta\sqrt{1-k/M}}\left(\beta + 2D\sqrt{\frac{2\pi}{N}}\left(1 + \sqrt{\frac{M}{k}}\right)\right).$$

$\square$

To analyze the convergence of Algorithm 4, we first need a bound on the expected norm of the momentum tensors $\mathbf{M}_t^i$. The following lemma provides this bound under Assumption 6.

**Lemma 1** (Bound on momentum $\mathbf{M}_t^i$). *Suppose Assumption 6 holds, and let $\{\mathbf{M}_t^i\}_{t=1}^T$ be the momentum tensor generated by Algorithm 4. Then, for all $t \geq 0$, we have the following bound on its expected norm:*

$$\mathbb{E}[\|\mathbf{M}_t^i\|_2] \leq \frac{R\beta\sqrt{1-k/M}}{1-\beta\sqrt{1-k/M}},$$

*provided that $\beta\sqrt{1-k/M} < 1$.*

*Proof.* Recall from Lemma 8.1 that the momentum $\mathbf{M}_t^i$ follows the recursion:

$$\|\mathbf{M}_t^i\|_2 \leq \sqrt{1-\frac{k}{M}}\|\beta\mathbf{M}_{t-1}^i + \mathbf{G}_t^i\|_2.$$

Taking expectations and applying the triangle inequality, we have:

$$\mathbb{E}[\|\mathbf{M}_t^i\|_2] \leq \sqrt{1-\frac{k}{M}}\left(\beta\,\mathbb{E}[\|\mathbf{M}_{t-1}^i\|_2] + \mathbb{E}[\|\mathbf{G}_t^i\|_2]\right).$$

Iterating this bound recursively, we obtain:

$$\begin{aligned}
\mathbb{E}[\|\mathbf{M}_t^i\|_2] &\leq \sqrt{1-\frac{k}{M}}\left(\beta\,\mathbb{E}[\|\mathbf{M}_{t-1}^i\|_2] + R\right) \\
&\leq (1-\frac{k}{M})\beta^2\mathbb{E}[\|\mathbf{M}_{t-2}^i\|_2] + R\sqrt{1-\frac{k}{M}}\left(1+\beta\sqrt{1-\frac{k}{M}}\right) \\
&\leq \cdots \\
&\leq R\sqrt{1-\frac{k}{M}}\sum_{j=1}^t\left(\beta\sqrt{1-\frac{k}{M}}\right)^{t-j}.
\end{aligned}$$

The geometric series can be bounded as follows:

$$\sum_{j=1}^t\left(\beta\sqrt{1-\frac{k}{M}}\right)^{t-j} \leq \frac{R\sqrt{1-k/M}}{1-\beta\sqrt{1-k/M}},$$

assuming $\beta\sqrt{1-k/M} < 1$ for convergence.

Thus, we have the final bound:

$$\mathbb{E}[\|\mathbf{M}_t^i\|_2] \leq \frac{R\sqrt{1-k/M}}{1-\beta\sqrt{1-k/M}},$$

completing the proof. □

The analysis of sparse top-$k$ averaging begins with Lemma 2, which provides a bound on the expected bias discrepancy $\mathbb{E}[\delta_i]$, highlighting the trade-offs between sample size, dimension, and sparsity. This is complemented by Lemma 8.1, which establishes a universal bound for the $\ell_1$-norm approximation error, illustrating vector approximation through top-$k$ component selection.

Furthermore, Lemma 8.3 offers insights into the geometric relationships between vectors and their sign functions, connecting inner product space and $L_1$-norm differences. This is supported by Lemmas 8.2, which explore sign-based approximation properties and variance.

**Lemma 2** (Bias Bound for Sparse Top-$k$ Averaging). *Let integers $N, M, k$ satisfy $1 \leq k \leq M$, and let random vectors $\{X^{(j)}\}_{j=1}^N \subseteq \mathbb{R}^M$ be independent and identically distributed, explicitly satisfying:*

*(i) **Coordinate independence:** For each $j \in \{1, \ldots, N\}$, the coordinates $\{X_i^{(j)}\}_{i=1}^M$ are independent and identically distributed random variables:*

$$X_i^{(j)} \overset{i.i.d.}{\sim} X_i, \quad and \quad \mathbb{E}[X_i] = \mu_i.$$

(ii) **Boundedness:** *There exists a finite constant $B > 0$ such that, almost surely,*

$$|X_i^{(j)}| \leq B, \quad \text{for all } i \in \{1, \ldots, M\}, \ j \in \{1, \ldots, N\}.$$

*Consider the estimator $\bar{X} \in \mathbb{R}^M$ defined by Algorithm 3 (Sparse Top-k Averaging). Define the standard empirical mean as*

$$\bar{X}^{(\text{full})} := \frac{1}{N} \sum_{j=1}^{N} X^{(j)},$$

*and the bias discrepancy introduced by sparsity as, for each coordinate $i \in \{1, \ldots, M\}$,*

$$\delta_i := |\bar{X}_i - \bar{X}_i^{(\text{full})}|.$$

*Then, under these assumptions, there exists a universal constant $C > 0$, independent of $N, M, k$, and $B$, such that for every coordinate $i \in \{1, \ldots, M\}$,*

$$\mathbb{E}[\delta_i] \leq C B \sqrt{\frac{1}{N}} \left( 1 + \sqrt{\frac{M}{k}} \right).$$

*Proof.* Recall that

$$\delta_i = |\bar{X}_i^{(k)} - \bar{X}_i^{(M)}| \leq \left| \frac{1}{n_i} \sum_{j \in \mathcal{J}_i} X_i^j - \mu_i \right| + \left| \frac{1}{N} \sum_{j=1}^{N} X_i^j - \mu_i \right| := T_1 + T_2.$$

**Bounding the first term $T_1$.**

The variable $n_i$ follows a Binomial$(N, p_i)$ distribution with $p_i = k/M$. Conditional on $n_i$, Hoeffding's inequality gives

$$P\left(T_1 \geq \eta \mid n_i\right) \leq 2 \exp\left(-\frac{2 n_i \eta^2}{B^2}\right), \quad n_i > 0.$$

We split the analysis into two cases based on $n_i$:

*Case 1: Typical case ($n_i \geq N p_i / 2$).* Using Hoeffding's inequality, we bound

$$\mathbb{E}[T_1 \mid n_i \geq N p_i / 2] \leq C'' B \sqrt{\frac{1}{N p_i}} = C'' B \sqrt{\frac{M}{N k}},$$

for some absolute constant $C'' > 0$.

*Case 2: Rare case ($n_i < N p_i / 2$).* Chernoff bounds for binomial distributions imply

$$P(n_i < N p_i / 2) \leq \exp(-c N p_i),$$

for some absolute constant $c > 0$. Thus, we trivially bound $T_1 \leq 2B$ to obtain

$$\mathbb{E}[T_1 \mid n_i < N p_i / 2] \leq 2B \exp(-c N p_i),$$

which becomes negligible for large $N$ relative to the polynomial terms.

Combining both cases, we have for large $N$,

$$\mathbb{E}[T_1] \leq C'' B \sqrt{\frac{M}{N k}} + o\left(\frac{1}{\sqrt{N}}\right).$$

**Bounding the second term $T_2$.**

Since $X_i^j$ are bounded in $[-B, B]$, Hoeffding's inequality gives, for any $\eta > 0$,

$$P\left(T_2 \geq \eta\right) \leq 2 \exp\left(-\frac{2 N \eta^2}{B^2}\right).$$

Integrating over all $\eta$ yields

$$\mathbb{E}[T_2] \leq \int_0^\infty 2 \exp\left(-\frac{2Nt^2}{B^2}\right) dt = C'B\frac{1}{\sqrt{N}},$$

for some absolute constant $C' > 0$.

**Combining both terms $T_1$ and $T_2$.**

Summing the bounds for $T_1$ and $T_2$, we obtain for sufficiently large $N$:

$$\mathbb{E}[\delta_i] \leq C'B\frac{1}{\sqrt{N}} + C''B\sqrt{\frac{M}{Nk}} + o\left(\frac{1}{\sqrt{N}}\right).$$

Factoring $B/\sqrt{N}$ explicitly, we conclude

$$\mathbb{E}[\delta_i] \leq CB\sqrt{\frac{1}{N}}\left(1 + \sqrt{\frac{M}{k}}\right),$$

for an absolute constant $C$ large enough (e.g., $C = 2\max(C', C'')$). This completes the proof. $\square$

**Remark 2** (Dependence on Parameters). *The bound in Lemma 2 clearly illustrates the scaling of the bias with respect to the number of samples $N$, dimension $M$, sparsity level $k$, and coordinate magnitude bound $B$:*

- ***Sample size** ($N$): The bias decreases at a rate of $1/\sqrt{N}$, consistent with standard statistical estimation theory.*

- ***Dimension and sparsity** ($M, k$): The bias scales as $\sqrt{M/k}$, highlighting the bias introduced by aggressive sparsification (small $k$).*

- ***No sparsification** ($k = M$): In this case, the bias vanishes, as expected.*

Here we present and prove a bound concerning the approximation error incurred by approximating a vector by its top-$k$ largest magnitude components.

**Lemma 8.1** (Optimal Universal $\ell_1$-norm Bound). *Let $X = (X_1, X_2, \ldots, X_M) \in \mathbb{R}^M$ be an arbitrary real vector. For a fixed integer $k$ with $1 \leq k \leq M$, define the vector $X^* \in \mathbb{R}^M$ by selecting the $k$ largest magnitude elements of $X$ and setting all other entries to zero. Then, the approximation error satisfies the bound*

$$\|X^* - X\|_1 \leq \left(1 - \frac{k}{M}\right)\|X\|_1.$$

*Furthermore, this bound is sharp. Equality holds if and only if the magnitudes of all elements of $X$ are equal.*

*Proof of Lemma 8.1.*
**Notation and Problem Setup.** Consider the vector $X \in \mathbb{R}^M$. Let us sort the elements of $X$ in non-increasing order of absolute value:

$$|X|_{(1)} \geq |X|_{(2)} \geq \cdots \geq |X|_{(M)}.$$

We define $X^*$ by retaining the $k$ largest magnitude elements and setting the remaining $M - k$ elements to zero. Thus, the approximation error is explicitly given by:

$$\|X^* - X\|_1 = \sum_{i=k+1}^M |X|_{(i)},$$

while the original vector norm is:

$$\|X\|_1 = \sum_{i=1}^M |X|_{(i)}.$$

**Identifying the Worst-case Scenario.** To determine the maximal possible ratio of the approximation error to the original norm, consider the scenario where all vector entries are equal in magnitude. Suppose, without loss of generality, that:

$$X = (a, a, \ldots, a), \quad a \neq 0.$$

In this case, any selection of $k$ elements yields the same approximation error. Explicitly, we have:

$$\|X^* - X\|_1 = (M - k)|a|, \quad \|X\|_1 = M|a|.$$

Hence, the ratio is exactly:

$$\frac{\|X^* - X\|_1}{\|X\|_1} = \frac{M - k}{M} = 1 - \frac{k}{M}.$$

**Universality and Optimality of the Bound.** We now verify that this equal-magnitude scenario indeed represents the worst-case for any vector $X$. For an arbitrary vector $X$, the approximation error ratio is:

$$\frac{\sum_{i=k+1}^{M} |X|_{(i)}}{\sum_{i=1}^{M} |X|_{(i)}}.$$

This ratio is maximized precisely when the magnitudes of vector elements are equal. To see this, observe that any deviation from equal magnitudes increases the share of the total magnitude held by the top-$k$ elements, thus reducing the ratio. Hence, the equal magnitude scenario is indeed the worst-case.

Therefore, the universal bound:

$$\|X^* - X\|_1 \leq \left(1 - \frac{k}{M}\right) \|X\|_1$$

holds for all vectors $X \in \mathbb{R}^M$. □

**Remark 3.** *Similarly, for $\ell - 2$ norm, we have*

$$\|X^* - X\|_2 \leq \sqrt{1 - \frac{k}{M}} \|X\|_2$$

**Remark 4** (Approximation error bound under isometric transformations). *Let $\mathcal{A} : \mathbb{R}^M \to \mathbb{R}^M$ be an invertible linear operator satisfying the isometric property*

$$\|\mathcal{A}(X)\|_2 = \|X\|_2, \quad \|\mathcal{A}^{-1}(X)\|_2 = \|X\|_2, \quad \text{for all } X \in \mathbb{R}^M.$$

*Define the vector approximation*

$$X^* = \mathcal{A}^{-1}\big(\text{top}_k(\mathcal{A}(X))\big),$$

*where $\text{top}_k(Y)$ retains only the $k$ largest-magnitude entries of $Y$ and sets the others to zero.*

*Then the approximation error satisfies the bound*

$$\|X^* - X\|_2 \leq \sqrt{1 - \frac{k}{M}} \|X\|_2.$$

*This bound is sharp, with equality attained precisely when the entries of $\mathcal{A}(X)$ have identical magnitudes. In particular, when $k = M$, the approximation error is exactly zero, indicating perfect reconstruction.*

**Lemma 8.2.** *For any $x, y \in \mathbb{R}$, we have*

$$|x| - x \operatorname{sign}(y) \leq 2|x - y|.$$

*Proof.* If $\operatorname{sign}(y) = \operatorname{sign}(x)$, we have $|x| - x \operatorname{sign}(y) = 0 \leq 2|x - y|$.

If $\operatorname{sign}(y) = -\operatorname{sign}(x)$, we have $|x| - x \operatorname{sign}(y) = 2|x| \leq 2|x| + 2|y| = 2|x - y|$.

If $\operatorname{sign}(y) = 0$, we have $|x| - x \operatorname{sign}(y) = |x| = |x - y| \leq 2|x - y|$. □

---

**Algorithm 5** Decoupled Momentum – High-level Training Loop

---

**Require:** learning rate $\eta$, momentum $\beta$, weight decay $\lambda$, sparsity $k$, DCT matrices $\{P_\ell\}$
1: Initialise parameters $\mathbf{X}_0$ and global momentum $\mathbf{M}_0 \leftarrow \mathbf{0}$
2: **for** $t = 1, 2, \ldots$ **do**
3:     **// executed on every worker $i$ (in parallel)**
4:     $\mathbf{G}_t^i \leftarrow \nabla\mathcal{L}(\mathbf{X}_{t-1}; \xi_t^i)$
5:     $\mathbf{M}_t^i \leftarrow \beta\,\mathbf{M}_{t-1}^i + \mathbf{G}_t^i$
6:     $\mathbf{Q}_t^i \leftarrow \text{COMPRESSCHUNKS}(\mathbf{M}_t^i, k, \{P_\ell\})$
7:     **send $\mathbf{Q}_t^i$ to server**
8:     **// parameter server**
9:     $\mathbf{M}_t \leftarrow \text{DECOMPRESS\&AGGREGATE}\big(\{\mathbf{Q}_t^i\}_i, \{P_\ell^\top\}\big)$
10:    $\mathbf{X}_t \leftarrow \mathbf{X}_{t-1} - \eta\big(\text{sign}(\mathbf{M}_t) + \lambda\,\mathbf{X}_{t-1}\big)$
11:    **broadcast** $\text{sign}(\mathbf{M}_t)$ **to all workers**
12: **end for**

---

**Algorithm 6** COMPRESSCHUNKS($\mathbf{M}, k, \{P_\ell\}$) — run on one worker

---

**Require:** local momentum $\mathbf{M}$, sparsity budget $k$, DCT matrices $\{P_\ell\}$
1: $\mathbf{Q} \leftarrow \emptyset$ {list of sparse coefficients}
2: **for** each chunk index $c$ **do**
3:     $B \leftarrow \text{CHUNKEXTRACT}(\mathbf{M}, c)$
4:     $\mathbf{q} \leftarrow \text{DCT}(B; \{P_\ell\})$
5:     $\mathbf{q}_{\text{sp}} \leftarrow \text{Top-}k(\mathbf{q}, k)$
6:     $\mathbf{Q} \leftarrow \mathbf{Q} \cup \{(c, \mathbf{q}_{\text{sp}})\}$
7:     $B \leftarrow B - \text{IDCT}(\mathbf{q}_{\text{sp}}; \{P_\ell^\top\})$ {store residual}
8:     $\text{CHUNKINSERT}(\mathbf{M}, c, B)$
9: **end for**
10: **return** $\mathbf{Q}$ {sparse DCT coefficients to transmit}

---

**Lemma 8.3.** *Let $(X, Y)$ is a joint random variable on $\mathbb{R}^d \times \mathbb{R}^d$. For any constant $a \in (0, +\infty)$, we have*

$$\mathbb{E}[\langle X, \text{sign}(X) - \text{sign}(Y)\rangle] \leq 2a\sqrt{d}\,\mathbb{E}\|X/a - Y\|.$$

*Proof.* Without loss of generality, set $a = 1$.

$$\begin{aligned}
\mathbb{E}[\langle X, \text{sign}(X) - \text{sign}(Y)\rangle] &= \mathbb{E}[\|X\|_1 - \langle X, \text{sign}(Y)\rangle] \\
&\leq 2\mathbb{E}[\|X - Y\|_1] \quad \text{Lemma 8.2} \\
&\leq 2\sqrt{d}\,\mathbb{E}[\|X - Y\|] \quad \text{by Cauchy-Schwarz,}
\end{aligned}$$

where $\|\cdot\|_1$ is the $\ell_1$ norm and $\|\cdot\|$ denotes the Euclidean norm. $\qquad\square$

---

**Algorithm 7** DECOMPRESS&AGGREGATE($\{\mathbf{Q}^i\}, \{P_\ell^\top\}$) — server side

---

**Require:** sparse sets $\{\mathbf{Q}^i\}$ from all workers, inverse DCT matrices $\{P_\ell^\top\}$
1: initialise global momentum $\mathbf{M} \leftarrow \mathbf{0}$
2: **for** each chunk index $c$ **do**
3:    $\mathbf{s} \leftarrow \mathbf{0}$ {sum of sparse coeffs}
4:    **for** each worker $i$ **do**
5:       $\mathbf{s} \leftarrow \mathbf{s} + \text{LOOKUP}(\mathbf{Q}^i, c)$
6:    **end for**
7:    $B \leftarrow \text{IDCT}(\mathbf{s}; \{P_\ell^\top\})$
8:    CHUNKINSERT($\mathbf{M}, c, B$)
9: **end for**
10: **return** $\mathbf{M}$ {reconstructed global momentum}

---

| Model | Final Loss ↓ | Hellaswag ↑ acc_norm | ARC-Easy ↑ acc | PIQA ↑ acc_norm | Data Tx ↓ MB/step |
|---|---|---|---|---|---|
| **DeMo 300M** | | | | | |
| $s=64,\ k=32$ | 2.87 | 0.37 | 0.46 | 0.67 | 29.9 |
| $s=64,\ k=16$ | 2.87 | 0.38 | 0.50 | 0.67 | 14.9 |
| $s=64,\ k=8$ | 2.88 | 0.38 | 0.47 | 0.67 | 7.49 |
| $s=64,\ k=4$ | 2.89 | 0.37 | 0.47 | 0.67 | 3.74 |
| $s=64,\ k=2$ | 2.93 | 0.36 | 0.46 | 0.65 | 1.87 |
| $s=64,\ k=1$ | 2.97 | 0.35 | 0.45 | 0.65 | 0.93 |
| $s=128, k=32$ | 2.88 | 0.37 | 0.50 | 0.66 | 7.49 |
| $s=128, k=16$ | 2.90 | 0.37 | 0.47 | 0.67 | 3.74 |
| $s=128, k=8$ | 2.93 | 0.36 | 0.49 | 0.66 | 1.87 |
| $s=128, k=4$ | 2.98 | 0.35 | 0.46 | 0.64 | 0.93 |
| $s=128, k=2$ | 3.06 | 0.33 | 0.45 | 0.65 | 0.46 |
| $s=128, k=1$ | 3.16 | 0.31 | 0.45 | 0.63 | 0.23 |
| **AdamW-DDP 300M** | 2.98 | 0.35 | 0.46 | 0.65 | 636.9 |
| **DeMo 1B** | | | | | |
| $s=64,\ k=32$ | 2.63 | 0.48 | 0.55 | 0.70 | 110.32 |
| $s=64,\ k=16$ | 2.63 | 0.47 | 0.53 | 0.70 | 55.16 |
| $s=64,\ k=8$ | 2.64 | 0.47 | 0.52 | 0.69 | 27.58 |
| $s=64,\ k=4$ | 2.67 | 0.45 | 0.52 | 0.70 | 13.79 |
| $s=64,\ k=2$ | 2.71 | 0.44 | 0.51 | 0.69 | 6.89 |
| $s=64,\ k=1$ | 2.76 | 0.41 | 0.52 | 0.69 | 3.44 |
| $s=128, k=32$ | 2.65 | 0.46 | 0.53 | 0.69 | 27.58 |
| $s=128, k=16$ | 2.67 | 0.46 | 0.50 | 0.70 | 13.79 |
| $s=128, k=8$ | 2.72 | 0.44 | 0.52 | 0.68 | 6.89 |
| $s=128, k=4$ | 2.76 | 0.41 | 0.50 | 0.67 | 3.44 |
| **AdamW-DDP 1B** | 2.73 | 0.43 | 0.51 | 0.68 | 2416.6 |

Table 3: Results of training loss, downstream evaluation scores, and per-GPU communication requirements of the model sizes and reference trained on 100B tokens

