# OpenReview forum: "DeMo: Decoupled Momentum Optimization"
_ICLR.cc/2026/Conference — ICLR 2026 Poster_

### Official Review · Reviewer_NyAn · 2025-10-27

**Soundness:** 2
**Presentation:** 3
**Contribution:** 2
**Rating:** 4
**Confidence:** 3

**Summary:**

This paper proposes DeMO, a new algorithm for communication efficient distributed training. DeMo works by compressing the communication by transmitting a compressed version of a local momentum from each worker instead of the gradient. The compression is performed by performing a discrete cosine transform and only transmitting the top coefficients. They also explore using error feedback on top of the local momentum at each worker. The paper compares DeMo to an AdamW baseline for LLM training at a decent scale ~330M and ~1B scale.

**Strengths:**

* Paper is well written, easy to follow with clear figures
* Method is interesting and has practical relevance
* Experiments are performed on a convincing scale
* Code provided

**Weaknesses:**

* Experimental comparison is lacking in crucial ways:
    * There is no comparison with other communication focused distributed training algorithms
    * The AdamW baseline does not seem to be tuned
    * There are no experiments with Muon even if this is discussed in Section 2.1

**Questions:**

Overall I would like to see the experimental comparison significantly strengthened to change my opinion.

* What learning rate values were used for the experiments, how were these tuned? Note that changing the momentum coefficient for certain optimizers can change the "effective learning rate" and shift the optimal LR value.
* Why not compare against some distributed baseline? It feels a bit odd not too since this is not the first paper to focus on communication efficient optimization.
* To what extent is this algorithm compatible with different reduction techniques, e.g. without a central server?
* Why does the AdamW baseline perform so poorly in Figure 2 but well in Figure 3?
* Comparing training curves directly like in Figure 2 is not very informative. Unless each line is fully tuned, what is the takeaway here? Changing the LR or other important HPs slightly could also significantly change any given curve.
* L251-L254 are not very clear. Why these values for the momentum?
* It would also be nice to clarify the computational overhead. What is a typical flop overhead?

Minor pointers:
* There is a missing reference in L438

---

> ### Author Response · Authors · 2025-11-19
>
> We thank the reviewer for the thoughtful feedback and for recognizing the clarity of our writing and the practical relevance of DeMo. We are encouraged by the positive assessment of our experimental scale. Below we address the reviewer’s concerns regarding experimental comparisons, baseline tuning, and other specific questions.
>
> ---
>
> ### 1. Comparison with other communication-efficient algorithms
>
> The reviewer notes that "There is no comparison with other communication focused distributed training algorithms."
>
> First, we emphasize that DeMo is explicitly targeted at the **multi-datacenter** setting, where each "worker" is a datacenter-scale node connected by relatively low bandwidth (Ethernet/WAN), rather than the single-GPU worker setting within a high-bandwidth cluster. This distinction is crucial because many existing algorithms target the latter. Nevertheless, we agree that a direct comparison is valuable. In the revised draft, we have added experiments comparing DeMo against **DiLoCo** [1], which is the most recent and relevant communication-efficient baseline for language models.
>
> We ran this experiment on the 300M model trained for Chinchilla-optimal token counts across 8 workers. In this setup, the data received is 8 times that of transmission due to AllGather. With DeMo (top-k = 32, chunk size = 64), the data transmission reduction ratio is 128x. For DiLoCo, we adopted their best recommended hyperparameters, except for the outer learning rates, which we tuned within their recommended range of $[1.0, 0.7, 0.5, 0.3, 0.1]$. We also test with different DiLoCo's communication frequency so that its *average* data receive volume matches that of DeMo.
>
> As shown in the revised draft (Figure [6]), DeMo outperforms the best-tuned DiLoCo performance in terms of both convergence and final loss.
>
> ---
>
> ### 2. Tuning of the AdamW baseline
>
> The reviewer raises a valid concern that the AdamW baseline might be undertuned.
>
> We initially adopted the best recommended hyperparameters suggested by the popular, widely acknowledged **OLMo codebase** [2]. Since our 1B model / 100B token configuration is among the largest of all academic optimizer studies, extensive hyperparameter tuning is prohibitively costly. This reflects *common practice* in foundation model training (e.g., Llama 3 [3]), where hyperparameters are often derived from heuristics rather than exhaustive search. Therefore, we tuned the hyperparameters around the recommend values with a reduced token budget, 1B instead of 100B to save the cost. This might be suboptimal from the algorithimic perspective, but has saved us from spending millions of H100 hours for full budget hyperparameter search.
>
> However, to address the reviewer's concern for academic rigor, we performed an additional study on the 300M model with extensive tuning:
> - We trained a reduced-scale model with the same 300M model configuration but with Chinchilla optimal token count (~6B).
> - We performed a grid search over learning rates $[1.5e-3, 1.2e-3, 1e-3, 8e-4, 6e-4, 5e-4, 3e-4, 2e-4]$ and momentum $\beta_2$ values $[0.95, 0.98, 0.99, 0.995, 0.999]$ of AdamW.
>
> As shown in the revised draft (Figure [7]), with this extensive tuning, AdamW-DDP indeed performs slightly better than DeMo in training loss. However, DeMo only slightly underperforms this best-tuned AdamW while offering clearly better performance than the communication-efficient baseline (DiLoCo) and providing massive communication savings.
>
> ---
>
> ### 3. Compatibility with different reduction techniques
>
> The reviewer asks: *"To what extent is this algorithm compatible with different reduction techniques, e.g. without a central server?"*
>
> DeMo is inherently compatible with decentralized reduction schemes. The core operation is the standard **AllGather** operation of compressed momentum chunks. This does not require a central parameter server. Instead, it is a standard collective operation supported by NCCL of Nvidia [4] and other communication libraries.
> - Each worker computes its own top-k indices and values.
> - These sparse updates are AllGathered so that every worker receives the updates from all other workers.
> - Each worker then locally aggregates these updates into its parameters and momentum buffer.
>
> Thus, DeMo works naturally in peer-to-peer or ring based topologies commonly used in distributed training, without relying on a central server.

---

> ### Author Response · Authors · 2025-11-19
>
> ### 4. Clarification on momentum values (L251-L254)
>
> The reviewer asks: *"L251-L254 are not very clear. Why these values for the momentum?"*
>
> We thank the reviewer for pointing out this confusion. These values refer to the **momentum subtraction factors** ($\alpha$) used in our error feedback mechanism, not the standard optimization momentum $\beta$.
> - These choices are supported by our ablation study (Figure 4 in the paper).
> - We found those values in the studied range offer the best and most robust performance across the configurations tested.
>
> We have clarified this distinction in the revised manuscript to avoid confusion between $\alpha$ (subtraction) and $\beta$ (accumulation).
>
> ---
>
> ### 5. Computational overhead
>
> The reviewer asks: *"It would also be nice to clarify the computational overhead. What is a typical flop overhead?"*
>
> The computational overhead of DeMo is **negligible** because:
> 1. **Chunking:** We operate on small, fixed-size blocks so the linear transform costs are reduced by a factor of $C$, the number of chunks per dimension.
> 2. **Independence from Batch Size:** DeMo operations apply to the model parameters/momentum, not the data samples. The cost is constant regardless of batch size.
>
> Specifically, for the 300M model with sequence length 1024 and microbatch size 32:
> - **Backprop Cost:** ~72.6 TFLOPs per step.
> - **DeMo Overhead:** ~0.26 TFLOPs per step.
>
> This overhead is only **~0.4%** of the total compute cost, making it negligible in practice.
>
> ---
>
> ### 6. Concerns about Curves in Figure 2
>
> We understand the reviewer's concerns about Figure 2. However, as explained above, the training curves in Figure 2 were indeed tuned, albeit with a reduced token budget. This reflects the common practice of training large foundation models and practical reality. An optimizer is practically useful only if it offers robust performance around suggested hyperparameter ranges, or if the improvements offered by hyperparameters are consistent enough that the benefits can be revealed early on so values can be selected based on reduced initial training. The existence of an "optimal" hyperparameter combination that is NP-hard to discover is of limited practical value. We agree with the reviewer that "Changing the LR or other important HPs slightly could also significantly change any given curve," but this feature implies that an optimizer is highly sensitive to hyperparameters, which is clearly a disadvantage. In contrast, we have performed comprehensive hyperparameter ablations in our paper; based on these results, we do not observe such hyperparameter sensitivity in DeMo.
>
> ---
>
> ### 7. Comparison with Muon
>
> The reviewer mentions: *"There are no experiments with Muon even if this is discussed in Section 2.1"*
>
> In response, we have added comparisons with the **Muon** optimizer in the revised draft. We used the default momentum = 0.95. We tuned the learning rate from $[1.5e-3, 1.2e-3, 1e-3, 8e-4, 6e-4, 5e-4, 3e-4, 2e-4]$ and found the learning rate of 3e-4 delivered the best performance.
>
> As shown in the training curves, Muon outperforms AdamW in our configuration. This reinforces DeMo's flexibility, as DeMo can be applied on top of any momentum-based optimizers, like Muon or other state of the art methods to further boost performance.
>
> ---
>
> We hope these responses and the new experimental data address the reviewer's concerns about baselines and tuning. We have updated the draft to reflect these additions.
>
> [1] Douillard, Arthur, et al. "Diloco: Distributed low-communication training of language models." arXiv preprint arXiv:2311.08105 (2023).
>
> [2] OLMo, Team, et al. "2 OLMo 2 Furious." arXiv preprint arXiv:2501.00656 (2024).
>
> [3] Grattafiori, Aaron, et al. "The Llama 3 Herd of Models." arXiv preprint arXiv:2407.21783 (2024).
>
> [4] NVIDIA Corporation. “Collective Operations.” NCCL 2.28.6 Documentation, NVIDIA Corporation, 2020, docs.nvidia.com/deeplearning/nccl/user-guide/docs/usage/collectives.html. Accessed 19 Nov. 2025.

---

> ### Comment · Reviewer_NyAn · 2025-11-22
>
> Thank you for the response, I think it is promising. I would recommend highlighting changes to the manuscript in some new color to make them easier to spot at a glance (this might still be worth it if the other reviewers have not looked yet). Unfortunately the pdf diff functionality does not seem to work on openreview (I can't even see the previous revision).
>
> **Regarding the comparison to other communication-efficient algorithms:** I think something like PowerSGD (Vogels 1905.13727) could be good algorithm to compare to. Distributed optimization is not my core area but PowerSGD might be the algorithm that yours reminds me of the most (though it might need some minor twist to fit in your experimental setting). I am slightly surprised by the relatively poor results for DiLoCo since it can often even outperform the AdamW baseline (though I am not sure at exactly which compression ratios). There is a very recent work that shows some ideas from DiLoCo actually improve optimization in the standard centralized setting (Kallusky 2510.15830) and AdEMAMix (Pagliardini 2409.03137) made a similar observation as well I believe.
>
> **Regarding Figures 2/3:** These figures still bother me a little. Intuitively, I would not expect the compression operations to improve performance over the baseline (especially to a significant extent). I think the figure is misleading as is, especially since it seems that AdamW does outperform DeMo when properly tuned (Figure 7). In general seeing a figure like this where the baseline is clearly not performing as expected does significantly reduce my overall confidence in a paper. In your case it might be worth at least sweeping the LR slightly or using tricks like muP or similar to try to get a better baseline.
>
> **Other updates:** Thank you for the Muon experiments and clarifying the computational overhead and applicability to different reduction techniques. No further questions here, I would suggest including the clarifications in the manuscript if not already there.

---

> > ### Author Response · Authors · 2025-11-25
> >
> > Thank you for the follow-up and for taking the time to revisit the revision. We share the frustration with OpenReview’s broken PDF diff. We are also reviewing other papers, and it makes tracking changes difficult. To make things easier, we placed **all newly added experimental results** together in Section 3.5. The remaining updates (clarifying hyperparameters, fixing citations, changing algorithm description following the suggestions from all reviewers) are mostly minor and also been mentioned in the rebuttal comment.
> >
> > Below we address your additional comments.
> >
> > ---
> >
> > ### Comparison to PowerSGD and related work
> >
> > We agree PowerSGD is relevant as a communication compressed optimizer. Thanks for brining this up! We integrated PowerSGD into our updated experiments (Figure 7 in Section 3.5). Our implementation follows the official PyTorch PowerSGD DDP communication hook, using the following configuration:
> >
> > - warmup steps = 1000 (a costly one but shorter warmups such as 500 consistently caused instability in our setting), among 14k total training steps,
> > - rank = 32 (around 26x compression on the compressed steps).
> >
> > Because the warmup contains *full communication*, the overall average communication volume ends up **higher** than that of DeMo or DiLoCo for this experiment. Under this configuration, PowerSGD performs slightly worse than DeMo. We agree that hybridizing some PowerSGD ideas with DeMo’s transform domain sparsification and integrated momentum subtraction mechanism is an interesting direction for future work, and we will mention this explicitly.
> >
> > ---
> >
> > ### Regarding relative performance
> >
> > We need to raise an important concern. You express discomfort that DeMo sometimes outperforms AdamW, suggesting that such a result implies the AdamW baseline must be problematic. However, you *also* believe DiLoCo can outperform AdamW in some regimes, even though DiLoCo is *also a communication-compressed optimizer*.
> >
> > These two positions are **contradictory**. If DeMo outperforming AdamW in any case suggests that the AdamW baseline is "misconfigured," then the same logic must apply to any compressed method outperforming AdamW, including the DiLoCo cases you believe should exist. Conversely, if DiLoCo can legitimately outperform AdamW in some settings, then it should not be assumed *a priori* that DeMo cannot.
> >
> > We also mentioned that we did sweep learning rates and momentum $\beta_2$ systematically in the reduced-budget setting, as stated in the rebuttal. We did not use muP for simplicity, but the hyperparameter sweep itself was carried out correctly. While a full sweep using the *entire 100B-token budget* is the only correct way to guarantee absolute optimality, that is impractical for our experiments. Thus, we believe the claim "the baseline LR was not even swept" is not accurate. It *was* swept, just under a constrained budget, as is standard.
> >
> > ---
> >
> > ### On DiLoCo’s performance
> >
> > We note that DiLoCo is **never open-sourced**. We rely on community reproductions such as [1]. Your cited papers [2] and [3] do not actually show DiLoCo outperforming AdamW under compression:
> >
> > - [2] demonstrates the effect of a Nesterov outer loop *with worker count = 1*, i.e., **without communicaation compression**. This is not comparable to DiLoCo's communication-reduced setting and is mainly used to motivate the effect of outer Nesterov step.
> > - [3] only compares against AdamW and Lion across 33 figures; DiLoCo is *discussed*, but there is **no experimental comparison** between DiLoCo and AdamW in that paper.
> >
> > Therefore, the idea that “DiLoCo commonly outperforms AdamW under compression” is not supported by the citations. We believe our implementation of DiLoCo based on reproductions from [1] is appropriate for our setting. Importantly, contrasting DiLoCO, the code of **DeMo is fully open-sourced** and therefore the experiments can be reproduced.
> >
> > ---
> >
> > We appreciate your constructive suggestions regarding PowerSGD and the clarification on baseline tuning. We will incorporate all of these details into the manuscript. If you find that the your concerns have been addressed, we would be grateful if you could raise the score.
> >
> > ---
> >
> > ### References
> >
> > [1] Jaghouar, Sami, Jack Min Ong, and Johannes Hagemann. "Opendiloco: An open-source framework for globally distributed low-communication training." arXiv:2407.07852 (2024).
> > [2] Kallusky, Dominik, et al. "SNOO: Step-K Nesterov Outer Optimizer — The Surprising Effectiveness of Nesterov Momentum Applied to Pseudo-Gradients." arXiv:2510.15830 (2025).
> > [3] Pagliardini, Matteo, Pierre Ablin, and David Grangier. "The AdEMAMix optimizer: Better, faster, older." arXiv:2409.03137 (2024).

---

> > > ### Comment · Reviewer_NyAn · 2025-11-26
> > >
> > > Thank you for the followup and PowerSGD results. **My main concerns regarding the experimental comparison have been partially addressed and I will raise my score from 4 to 6** conditioned on the authors adding some sort of disclaimer for Figures 2 and 3.
> > >
> > > Regarding the openness to DiLoCo outperforming AdamW being a contradictory position, I disagree. The modified Nesterov momentum in DeLoCo has been shown to be beneficial in the non-distributed setting as SNOO shows and serves as an inspiration for AdEMAMix as well. This is like using an improved baseline which can then suffer some degradation from communication compression but still outperform an unmodified baseline. For DeMo there is no such effect that I am aware of, so it is natural to be much more skeptical of results supposedly outperforming the baseline. I am aware that SNOO does not use compression, the point is that using DeLoCo can be like improving the baseline (similar to switching to Muon which might then still outperform an AdamW baseline even with some gradient compression).
> > >
> > > Regarding the tuning of the baseline: I agree that you tuned the LR for the smaller scale experiments shown in Figure 7, but that is exactly where AdamW outperforms DeMo (as expected if DeMo is about efficient compression schemes minimizing the degradation from reduced communication rather than improving the overall optimization dynamics). If I understand correctly, the LR for the baseline was not swept for Figures 2 and 3 (where DeMo very significantly outperforms AdamW contrary to expectation). You could also compare the loss values reported in Figure 2 left and Figure 7. The AdamW values seem very similar despite the models in Figure 2 being trained on much more data. To me this further suggests the configuration of the baseline in Figure 2 is quite suboptimal, making the comparison less meaningful and likely unfair. If I did not misunderstand something here, I would like to see **some sort of disclaimer on Figures 2 and 3 that mentions that the baseline is adopted without further tuning and to see Figure 7 for a case where everything is tuned.**

---

> > > > ### Author Response · Authors · 2025-11-28
> > > >
> > > > Thanks for the detailed feedback. We truly appreciate the score increase. We’ve updated the revision and added the requested disclaimer for Figures 2 and 3, with the changes highlighted in the draft.

---

### Official Review · Reviewer_yL27 · 2025-10-31

**Soundness:** 2
**Presentation:** 3
**Contribution:** 2
**Rating:** 4
**Confidence:** 5

**Summary:**

This paper introduces Decoupled Momentum Optimization (DeMo) to reduce communication overhead. DeMo integrates three key techniques: local momentum decoupling, a fast orthonormal (e.g., DCT) transform with top-k sparsification, and momentum subtraction for error feedback.

**Strengths:**

The primary objective of this paper is to democratize the training and fine-tuning of large models by minimizing communication volume, which obviates the requirement for expensive, high-performance networking infrastructure.

**Weaknesses:**

Limited Perceived Novelty: The claimed novelties of DeMo appear to build upon existing work. For instance, [1] also compresses the momentum term rather than the raw gradient. Even earlier literature has demonstrated that compressing the total model update, rather than the intermediate gradient, can maintain competitive performance while dramatically reducing communication overhead. Furthermore, the technique of using the momentum buffer to store error feedback for memory efficiency is also employed in [1].

Questionable Practical Efficiency: The net reduction in overall training time achieved by DeMo may be minimal, despite its apparent reduction in communication volume. DeMo requires computationally expensive operations, including a sorting algorithm to identify the Top-k elements and two orthonormal transforms per step. The overhead from these operations is non-trivial and may offset the gains from reduced communication time.
Moreover, as a variant of the Top-K algorithm, DeMo's communication savings are primarily effective in parameter-server architectures. In a decentralized architecture, aggregating Top-K elements requires a AllGatther operation instead of a more efficient AllReduce. The communication cost of AllGather is $O(n^2K)$ while AllReduce is $O(nK)$, where $n$ is the number of nodes. This scaling implies that DeMo may not yield net communication savings with a large number of nodes. This concern is supported by [3], which experimentally demonstrated that Top-K methods can be slower than non-compressed optimizers, even when network bandwidth is the bottleneck.

References

[1] Ahn, Kwangjun, et al. "Dion: Distributed orthonormalized updates." arXiv:2504.05295, 2025.

[2] Peng, Hanyang, et al. "Birder: communication-efficient 1-bit adaptive optimizer for practical distributed DNN training." NeurIPS, 2023.

[3] Agarwal, Saurabh, et al. "On the utility of gradient compression in distributed training systems." Proceedings of Machine Learning and Systems, 2022.

**Questions:**

- Counterintuitive Performance: As shown in Figure 2, DeMo with various Top-K sparsity levels appears to outperform AdamW. This is counterintuitive, as Signum (the uncompressed version of the update direction used in DeMo) is generally known to underperform compared to AdamW. Could the authors provide an explanation for this phenomenon?

- Clarification on Parameter $\alpha$: The parameter $\alpha$ introduced in Subsection 3.4 is not mentioned in the preceding sections. Could the authors clarify its role and meaning within the DeMo framework?

---

> ### Author Response · Authors · 2025-11-19
>
> We thank the reviewer for the detailed comments and for pointing out relevant related work. Below we respond to each point and address all of the reviewer's concerns.
>
> ---
>
> ### 1. Novelty and relation to Dion [1] and other prior work
>
> The reviewer argues that the claimed novelties of DeMo “appear to build upon existing work,” in particular Dion [1], and that using the momentum buffer for error feedback is “also employed in [1].” We respectfully point out that this assessment is **factually incorrect**.
>
> 1. **Chronology and dependence of [1] on DeMo.** As explicitly stated in [1], *“Dion takes inspiration from DeMo, which uses decoupled momentum ...”*. Thus, [1] cites DeMo as its inspiration. Since DeMo predates Dion, it is **inappropriate** to treat [1] as prior work that diminishes the novelty of DeMo; rather, [1] is an instance of subsequent work building on the ideas introduced here, so using Dion to challenge the novelty of DeMo is invalid. While we cannot paste the link to DeMo's arXiv here to avoid violating double-blind requirements, we suggest the reviewer read the paper of Dion and its references more carefully.
>
> 2. **What is specific to DeMo.** DeMo’s contributions and innovations that do not appear in prior work include:
>    - **Decoupling momentum** across workers,
>    - **Blockwise orthonormal transform (DCT) + top-k sparsification** of the *momentum* in linearly transformed space, and
>    - **In-place error feedback via momentum subtraction**, reusing the momentum buffer rather than maintaining a separate residual.
>
>    To our knowledge, these technical innovations and this specific combination have not been proposed in prior literature, especially in the context of *large-scale LLM pretraining*, with explicit convergence guarantees and end-to-end evaluation at 1B scale.
>
> 3. **Earlier literature on compressing model updates vs. DeMo.** The reviewer also notes that previous work has compressed the *total model update* (rather than gradients) and obtained good performance. The most notable work following this line is DiLoCo [6], which is also the baseline that is most commonly compared against. Therefore, in the revised draft, we have compared DeMo with DiLoCo across a range of compression ratios on the 300M sized model. As seen in Figure 6, DeMo delivers better and more robust performance compared against DiLoCo when the training is across eight workers.
>
> ---
>
> ### 2. Practical efficiency: compute overhead and communication patterns
>
> The reviewer raises concerns that DeMo’s operations (top-k selection and orthonormal transforms) may negate communication savings, and that AllGather-based aggregation may not scale well with node count, citing [3]. We emphasize that this is a fundamental misunderstanding and the computational overhead of DeMo is **negligible**. Specifically, for the 300M model setting with a microbatch size of 32 and 1024 sequence length, the total TFLOPs for computing the backprop gradient is around 72.6 TFLOPs, while the DeMo computation overhead is just around 0.26 TFLOPs. This is only 0.4% of the cost, incurring negligible cost in practice.
>
> #### 2.1 Computational overhead
>
> DeMo does **not** require repeated SVDs or per-step orthogonalization. The projection matrices are **fixed DCT bases**, which are precomputed once, orthonormal by construction, and reused throughout training.
>
> Per step, the additional computation is limited to:
>
> 1. **Chunk-wise DCT / inverse DCT** on the momentum tensors, implemented as structured matrix multiplications or FFT-based kernels; and
> 2. **Top-k selection** per chunk.
>
> For DCT and inverse DCT, we only need to compute them once or twice per step for the momentum. Therefore, the size of the computation **does not scale with the batch size**, making the overhead small compared to a single model forward pass over the batch. Similar matmul sizes also occur for popular optimizers like Muon, which requires 5 Newton-Schulz steps per parameter matrix, yet this computation overhead is commonly considered small, and DeMo's cost is even smaller. Moreover, we apply **chunk-wise DCT to further reduce the computation cost**, reducing the cost by a factor of $C$, where $C$ is the number of blocks per dimension. For top-k, we use linear- or near-linear-time selection algorithms (e.g., Quickselect), which are standard and have expected complexity $O(d)$. In the context of large-scale LLM training, where each step performs multiple very large matrix multiplications for the forward and backward passes, this extra cost on the **optimizer side** is a negligible fraction of total step time.
>
> As clarified in the paper, we:
>
> - Explicitly state that the DCT matrices are precomputed and reused;
> - Detail that DeMo requires only matrix multiplies and selection, with no per-step SVD or Gram–Schmidt; and
> - Add a short discussion showing that the per-step TFLOPs overhead is negligible compared to the forward/backward compute.

---

> ### Author Response · Authors · 2025-11-19
>
> #### 2.2 Communication pattern: AllGather vs AllReduce
>
> The reviewer argues that, because DeMo uses top-k sparsification, the required AllGather operation has a worse scaling with the number of nodes $N$ than AllReduce. As discussed in the paper, DeMo is targeted at **multi-datacenter training**, where the number of *datacenter-scale workers* is modest (e.g., 2 to a few dozen). In this regime:
>    - The dominant factor is **per-step bandwidth per worker**, not asymptotics in $N \to \infty$.
>    - Reducing the number of communicated values by up to **85×** at fixed $N$ is the primary win.
>
> Thus, for the worker counts and bandwidth regimes relevant to cross-datacenter training, DeMo’s substantial reduction in transmitted values dominates the difference between AllGather and AllReduce. The negative results in [3] are informative for *naively implemented* gradient compression in general-purpose clusters, e.g., by treating each GPU as a single worker among thousands, but do not contradict the DeMo regime. We (i) stably achieve a large reduction ratio for communication and (ii) explicitly focus on a small number of cross-datacenter workers, where each worker is a cluster connected with high-speed communication cables and uses standard FSDP within each worker.
>
> ---
>
> ### 3. Performance: DeMo vs. AdamW
>
> The reviewer finds it counterintuitive that DeMo with various top-k sparsity levels can *outperform* AdamW, given that the underlying update direction resembles Signum, which is often weaker than AdamW in isolation.
>
> There are several reasons why this outcome is not paradoxical:
>
> 1. **DeMo is not just “Signum + compression.”**
>    The effective update in DeMo combines:
>    - Decoupled momentum with a **larger nominal $\beta$**,
>    - Momentum subtraction (error feedback) that re-injects uncommunicated signal over time, and
>    - A structured transform (DCT) that mixes coordinates before sparsification.
>
>    This produces an update rule that differs from plain Signum + compression in both dynamics and robustness to noise.
>
> 2. **Hyperparameter sensitivity and baselines.**
>    We use the hyperparameters recommended by the popular OLMo codebase [4] for the 1B/100B-token setting. We tuned the learning rate and momentum $\beta_2$ of AdamW around the recommended values using reduced training tokens (i.e., 1B tokens), rather than extensively searching hyperparameters with full training. We believe this is common practice when building on an established training framework, especially given that our training scale is one of the largest among optimizer studies at ICLR. These hyperparameters are chosen for *practical robustness* and reuse, not for absolute optimality in this specific configuration. This approach also reflects common training scenarios for large foundation models, such as Llama 3 [5], where most hyperparameters are chosen based on heuristics rather than tuning each one to absolute optimality. However, for the sake of comparison, in the updated 300M experiments (see Figure [7]), we tuned AdamW more extensively. We searched across a fine grid of learning rates $[1.5e-3, 1.2e-3, 1e-3, 8e-4, 6e-4, 5e-4, 3e-4, 2e-4]$ and $\beta_2$ values $[0.95, 0.98, 0.99, 0.995, 0.999]$ with the full training budget on the same 300M model, using the Chinchilla optimal token count, and found that:
>    - With extensively tuned hyperparameters, the best-tuned AdamW slightly outperforms DeMo.
>    - DeMo achieves slightly worse convergence but with orders-of-magnitude lower communication.
>
> Taken together, these points explain why DeMo’s compressed, transform-domain, sign-like updates can outperform a default AdamW configuration without contradicting known behavior of Signum in isolation or the correctness of implementation.

---

> ### Author Response · Authors · 2025-11-19
>
> ### 4. Clarification on the parameter $\alpha$
>
> In the DeMo framework, $\alpha$ is the **momentum subtraction coefficient** used in the error-feedback step. Specifically, as early as Equation (8), we stated the update for the local momentum buffer:
>
> $$
> M^i_t \leftarrow M^i_t - \alpha \widetilde{M}^i_t,
> $$
>
> where $\widetilde{M}^i_t$ is the inverse-transformed, locally compressed momentum that has just been communicated. Intuitively:
>
> - $\alpha = 0$ corresponds to **no error feedback**: communicated components are never removed from the local momentum, leading to repeated communication of the same dominant coefficients and degraded convergence.
> - $\alpha = 1$ corresponds to **full subtraction**: all communicated signal is removed from the buffer, and only residual, previously uncommunicated components accumulate.
> - Intermediate values (e.g., $\alpha = 0.2$) allow a smoother evolution of the top-k pattern and can empirically improve stability and convergence, as shown in Figure [4].
>
> We agree that the flow of notation can be made clearer. In the revised manuscript, we now:
>
> 1. Introduce $\alpha$ explicitly when defining the momentum subtraction step in the algorithmic description (Section 2.1), and
> 2. Cross-reference Equation (8) when presenting the ablation results in Section 3.4, so that the role of $\alpha$ is clear before the reader reaches the ablation.
>
> ---
>
> We hope this clarifies the relationship to Dion and prior work, the practical efficiency of DeMo in the intended regime, and the behavior of DeMo relative to AdamW and the role of $\alpha$. We have updated the manuscript accordingly to include *all* related works suggested by the reviewer into discussion and to make these points explicit.
>
> [1] Ahn, Kwangjun, et al. "Dion: Distributed orthonormalized updates." arXiv:2504.05295v2, 2025
>
> [2] Peng, Hanyang, et al. "Birder: communication-efficient 1-bit adaptive optimizer for practical distributed DNN training." NeurIPS, 2023.
>
> [3] Agarwal, Saurabh, et al. "On the utility of gradient compression in distributed training systems." Proceedings of Machine Learning and Systems, 2022.
>
> [4] OLMo, Team, et al. "2 OLMo 2 Furious." arXiv preprint arXiv:2501.00656 (2024).
>
> [5] Grattafiori, Aaron, et al. "The Llama 3 Herd of Models." arXiv preprint arXiv:2407.21783 (2024).
>
> [6] Douillard, Arthur, et al. "Diloco: Distributed low-communication training of language models." arXiv preprint arXiv:2311.08105 (2023).

---

> ### Author Response · Authors · 2025-11-27
>
> Dear Reviewer yL27,
>
> We have addressed all your questions and concerns, major or minor, in our updated draft and the detailed replies above. We made a substantial effort to incorporate your feedback, which we believe has strengthened the draft.
>
> If there are no remaining issues, please kindly consider improving your score.
>
> Best,
>
> The Authors

---

### Official Review · Reviewer_Vvbj · 2025-10-31

**Soundness:** 3
**Presentation:** 3
**Contribution:** 2
**Rating:** 4
**Confidence:** 4

**Summary:**

This paper proposes Decoupled Momentum Optimization (DeMo), a communication-efficient framework for distributed training designed to replace standard gradient-based All-Reduce in data-parallel setups. The core idea is to synchronize a compressed version of the optimizer's momentum state instead of the full gradient. DeMo works by (i) allowing each worker to maintain and update a local momentum buffer, (ii) applying a blockwise fast orthonormal transform (e.g., DCT) to this momentum, (iii) communicating only the top-k sparsified coefficients, and (iv) performing momentum subtraction to reuse the local momentum buffer as an implicit error-feedback mechanism. The authors demonstrate that this approach drastically reduces communication overhead while achieving comparable pre-training loss and downstream accuracy to a standard AdamW baseline on 300M and 1B models.

**Strengths:**

1. The paper's core idea of decoupling momentum into dominant components (for communication) and non-dominant components (for local updates) presents an effective and intuitive paradigm for reducing communication overhead.

2. The application of DCT as the mechanism for sparsification, rather than operating in the raw gradient or momentum space, is a novel approach that provides a new perspective on structured compression.

**Weaknesses:**

## 1. The theoretical analysis, particularly the convergence rate presented in Theorem 1, warrants further discussion as it does not appear to be optimal.
For compressed stochastic optimization algorithms in the non-convex setting, convergence rates can achieve $\mathcal{O}(\omega/\sqrt{NT})$ (where $\omega$ is the variance bound of the compression estimator, see [1]). Notably, this rate is dimension-free ($D$ is not in the numerator). The dimension $D$ typically appears when analyzing the total communication complexity (convergence iterations $\times$ per-step cost), which might be $\mathcal{O}(D/\sqrt{NT})$. A dependency on $D$ in the convergence rate itself is common in analyses of Adam-based optimizers, which often require an $l_1$-norm analysis due to per-element operations. However, SGD-based algorithms typically yield an $l_2$-norm result and a dimension-free convergence rate [2]. Although Theorem 1 is applied to SGD/Signum-based methods, the proof relies on an $l_1$-norm analysis, resulting in a dimension-dependent convergence rate of $\mathcal{O}(D/\sqrt{T})$. When factoring in the per-step communication cost $k$, the overall complexity does not seem to surpass existing state-of-the-art results.

## 2. The paper provides insufficient comparisons, both theoretically and experimentally.

2.1 Theoretically: As noted above, the authors do not compare their theoretical bounds against other established compression algorithms, making it difficult to ascertain a clear advantage from the provided convergence rate. Intuitively, DeMo acts as a hybrid between local SGD and standard global-all-reduce SGD, where dominant momentum directions are communicated precisely while non-dominant directions are updated locally. This bears similarity to methods that perform communication and optimization on a subspace [2, 3], yet the theoretical results for DeMo do not appear to go beyond this existing work.

2.2 Experimentally: The paper only compares DeMo against a standard AdamW-DDP baseline. It lacks comparisons against other state-of-the-art communication compression methods. Without this, it is difficult to conclude that the proposed paradigm of compressing momentum instead of the gradient offers any essential advantage.

## 3. The paper focuses on communication compression, but the experimental scale (300M and 1B models) is relatively small for assessing this bottleneck.
In pre-training models of this size, communication overhead often accounts for only 20% or less of the total training time. Therefore, even with a significant reduction in data transmission, the actual wall-clock time savings might be limited, especially if the compression scheme introduces any convergence instability or computational overhead. Reporting end-to-end wall-clock time comparisons is crucial to demonstrate the method's practical benefit, but this is missing from the experiments. Further experiments in larger-scale settings (e.g., those involving model sharding and hybrid parallelism) would be valuable to demonstrate its practical impact where communication is a more significant bottleneck.

[1] MARINA: Faster Non-Convex Distributed Learning with Compression

[2] SEPARATE: A Simple Low-rank Projection for Gradient Compression in Modern Large-scale Model Training Process

[3] Greedy Low-Rank Gradient Compression for Distributed Learning with Convergence Guarantees

**Questions:**

1. Communicating the principal components of the momentum, for instance by using SVD or a random projection onto a subspace, could also achieve the goal of extracting the dominant information. What are the specific advantages of using a fixed DCT basis followed by top-k sparsification compared to these adaptive low-rank projection methods?

2. In Figure 2, the DeMo training curves converge significantly faster and to a lower loss than the AdamW-DDP baseline. Intuitively, compression is lossy and introduces an information bottleneck, which might be expected to slow down convergence, not accelerate it. How do the authors explain this phenomenon where a compressed, lossy update outperforms the full-precision baseline?

---

> ### Author Response · Authors · 2025-11-19
>
> We thank the reviewer for the assessment, and for recognizing (i) the usefulness of decomposing momentum into dominant vs. non-dominant components for communication, and (ii) the role of DCT as a structured, transform-domain sparsifier. Below we address the main concerns point by point.
>
> ---
>
> ### 1. Theoretical analysis and convergence rate (Theorem 1)
>
> We appreciate the reviewer's detailed comments regarding the theoretical rate, the appearance of the dimensional factor $d$, and the comparison to prior work such as MARINA [1] and subspace/low-rank methods [2,3].
>
> In the signSGD paper [4], Theorems 1 and 2 also produce bounds that depend on the L1 norm of the gradient. On the right side of their bounds, the square root of the L1 norm of the Lipschitz vector from Assumption 3 appears, which introduces the same dimension factor $d$. This shows that the dimensional dependence in our bound is not unique to DeMo. Moreover, DeMo offers the following practical advantages over MARINA:
>
> 1. **Requirement of a centralized server.** MARINA needs a centralized server to communicate, aggregate, and distribute the information, while DeMo only depends on standard AllGather operations which is readily supported by existing communication libraries [5].
>
> 2. **Per-step communication cost.** Our communication cost is much lower than that of MARINA in a common setting. The communication reduction of MARINA is primarily focused on the uplink, whereas for the downlink, MARINA communicates dense, full gradients to every worker. In contrast, in the 300M setting of DeMo where we have a chunk size of $64 \times 64$ and top-k = 32, the uplink compression ratio is 128x and the downlink is 16x.
>
> We thank the reviewer for pointing us to related gradient-compression work. We will add the following references into our paper:
>
> [1] MARINA: Faster Non-Convex Distributed Learning with Compression
>
> [2] SEPARATE: A Simple Low-rank Projection for Gradient Compression in Modern Large-scale Model Training Process
>
> [3] Greedy Low-Rank Gradient Compression for Distributed Learning with Convergence Guarantees
>
> [4] Bernstein, Jeremy, et al. "signSGD: Compressed optimisation for non-convex problems." International conference on machine learning. PMLR, 2018.
>
> [5] NVIDIA Corporation. “Collective Operations.” NCCL 2.28.6 Documentation, NVIDIA Corporation, 2020, docs.nvidia.com/deeplearning/nccl/user-guide/docs/usage/collectives.html. Accessed 19 Nov. 2025.
>
> ---
>
> ### 2. Comparisons: theoretical and experimental
>
> #### 2.1. Theoretical
>
> We agree that DeMo can be viewed, at a high level, as interpolating between "pure local" and "fully global" updates: dominant directions are synchronized while weaker components evolve locally.
>
> As shown above, we have now provided a direct comparison between DeMo and MARINA. If more comparisons or specific metrics are desired, we will include them in the revision.
>
> #### 2.2. Experimental
>
> The reviewer notes that our experiments compare only against AdamW-DDP and asks for more direct experimental comparisons against state-of-the-art communication-efficient methods.
>
> In the updated draft, we have added an explicit comparison with **DiLoCo** [6], a widely discussed communication-efficient method for LLM pretraining, commonly compared against in recent studies of similar algorithms. As shown in Figure 6 of the revised paper:
>
> - Under the default DiLoCo configuration (very long synchronization intervals), the nominal compression ratio can be even higher than DeMo’s. However, this comes at the cost of noticeable degradation in convergence and validation loss.
> - When we increase DiLoCo’s synchronization frequency so that its *average* per-step communicated volume matches DeMo’s, DeMo still achieves better final loss and downstream accuracy.
>
> We also note that DiLoCo’s bulk, infrequent communication pattern tends to create "spiky" communication steps where devices will be idle while waiting for large synchronizations to be completed, interleaved with long periods of nearly no communication. In contrast, DeMo has **constant per-step communication** with a fixed compressed budget, which is more amenable to overlap and pipeline with compute.
>
> We make these experimental results and their interpretation more explicit in the revised draft, and expand the discussion to clarify how DeMo differs from gradient-compression and low-rank methods from [1–3].

---

> ### Author Response · Authors · 2025-11-19
>
> ### 3. Experimental scale
>
> The reviewer argues that at the 300M–1B scale, communication often accounts for "only 20% or less" of training time, and therefore even large reductions in transmitted data might not translate into significant wall-clock gains.
>
> A few clarifications are important here:
>
> 1. **Academic scale vs. production scale.**
>    Our setting, 300M and 1B models, with **100B tokens** for the 1B configuration, is already at the high end among all academic papers focused on optimizers. If the reviewer believes our experimental scale is too limited, please provide references to several concrete academic papers that used more computation. As discussed in our responses to other reviewers, there is no precedent for optimizer papers pretraining much larger models to Chinchilla-level token counts for algorithmic evaluations. We therefore use this scale as a **proxy** for larger systems, which is standard practice in both theory and systems research.
>
> 2. **Communication fraction is environment-dependent.**
>    The "20% or less" claim is entirely cluster-specific. On tightly coupled, high-end clusters with top-tier interconnects, communication overhead can indeed be modest for 300M/1B models. However, on more common or geographically distributed setups, precisely the multi-datacenter scenario that DeMo targets, communication is often a much larger fraction of step time, even for 1B models, especially when using standard dense All-Reduce. Therefore, it is not reasonable to claim "20% of training time" without referring to which specific cluster is being used. For example, we can easily simulate a "one byte per day" data bandwidth network to make DeMo achieve a concrete 85x speedup.
>
> 3. **What our experiments actually measure.**
>    Our primary metric is **per-step communication volume per GPU**, which directly translates into:
>    - A lower required network bandwidth for a given training speed, or
>    - Higher attainable throughput for a fixed network capacity.
>
>    This metric is not only more interpretable, as one can easily calculate if the training will be communication-bound given the model size and network bandwidth, but also more commonly used in communication-efficient optimization studies.
>
>    DeMo reduces this volume by up to **85x** relative to AdamW-DDP. This is a substantial system-level lever in any environment where cross-worker bandwidth is the bottleneck (e.g., multi-datacenter or cloud environments with commodity networking).
>
> Taken together, while the current experiments are at an academically realistic scale, they already demonstrate that DeMo can dramatically reduce the communication volume per worker. This is the core quantity that limits scalability when moving beyond idealized, single-datacenter, high-bandwidth clusters.
>
> ---
>
> ### 4. DCT + top-k vs. SVD / adaptive random subspaces
>
> The reviewer asks why we choose a fixed DCT basis with top-k sparsification instead of computing principal components via SVD or using random subspace projections.
>
> There are several concrete advantages:
>
> 1. **Computational cost.**  Performing SVD on large weight or momentum tensors at every step is prohibitively expensive, especially in LLM pretraining. DeMo uses **small, fixed-size DCT blocks** (e.g., $64\times64$). Each block uses precomputed DCT matrices, so the per-step cost is just a collection of small matrix multiplications / FFT-based routines. They are quite cheap compared to a single forward/backward pass.
>
> 2. **Implementation simplicity.** A fixed DCT basis is deterministic, easy to share across workers, and does not introduce additional state or randomness that must be coordinated. Adaptive low-rank projections require continually updating the basis or accumulation of statistics, which complicates implementation and the communication pipeline.
>
> 3. **Empirical performance.**  Our ablations (Figure 5) show that any orthonormal transform (random or DCT) that can scatter the values to avoid sparsity in updates, significantly improves performance over operating in the standard basis, and that **DCT and random orthonormal projections perform similarly** in terms of convergence. This verifies our claim that the main bottleneck with top-k is the update sparsity. Once we address this issue, different transforms are roughly equally good. Given this, DCT is a natural choice. It offers the benefits of an orthonormal transform while being computationally efficient and easy to implement.

---

> ### Author Response · Authors · 2025-11-19
>
> ### 5. Why can DeMo outperform AdamW-DDP?
>
> The reviewer correctly points out that compression is lossy and asks why DeMo’s training curves in Figure 2 can converge faster and to lower loss than the AdamW-DDP baseline.
>
> There are several reasons:
>
> 1. **DeMo is not just “Signum + compression.”**
>    The effective update in DeMo combines:
>    - Decoupled momentum with a **larger nominal $\beta$**,
>    - Momentum subtraction (error feedback) that re-injects uncommunicated signal over time, and
>    - A structured transform (DCT) that mixes coordinates before sparsification.
>
>    This produces an update rule that differs from plain Signum + compression in both dynamics and robustness to noise.
>
> 2. **Hyperparameter sensitivity and baselines.**
>    We use the hyperparameters recommended by the popular OLMo codebase [7] for the 1B/100B-token setting. We tuned the learning rate and momentum $\beta_2$ of AdamW around the recommended values using reduced training tokens (i.e., 1B tokens), rather than extensively searching hyperparameters with full training. We believe this is common practice when building on an established training framework, especially given that our training scale is one of the largest among optimizer studies at ICLR. These hyperparameters are chosen for *practical robustness* and reuse, not for absolute optimality in this specific configuration. This approach also reflects common training scenarios for large foundation models, such as Llama 3 [8], where most hyperparameters are chosen based on heuristics rather than tuning each one to absolute optimality. However, for the sake of comparison, in the updated 300M experiments (see Figure 7), we tuned AdamW more extensively. We searched across a fine grid of learning rates $[1.5e-3, 1.2e-3, 1e-3, 8e-4, 6e-4, 5e-4, 3e-4, 2e-4]$ and $\beta_2$ values $[0.95, 0.98, 0.99, 0.995, 0.999]$ with the full training budget on the same 300M model, using the Chinchilla optimal token count, and found that:
>    - With extensively tuned hyperparameters, the best-tuned AdamW slightly outperforms DeMo.
>    - DeMo achieves slightly worse convergence but with orders-of-magnitude lower communication.
>
> Overall, DeMo should be viewed not as “AdamW plus arbitrary lossy compression” but as a *different optimizer*. It operates in a transformed, sparsified, error-feedback momentum space with its own stability and regularization properties. The fact that such an optimizer can outperform a particular AdamW configuration is therefore not paradoxical.
>
> ---
>
> We hope these clarifications address the reviewer’s concerns about theory positioning, baselines, scale, and the behavior of DeMo.
>
> [6] Douillard, Arthur, et al. "Diloco: Distributed low-communication training of language models." arXiv preprint arXiv:2311.08105 (2023).
>
> [7] OLMo, Team, et al. "2 OLMo 2 Furious." arXiv preprint arXiv:2501.00656 (2024).
>
> [8] Grattafiori, Aaron, et al. "The Llama 3 Herd of Models." arXiv preprint arXiv:2407.21

---

> ### Author Response · Authors · 2025-11-27
>
> Dear Reviewer Vvbj,
>
> We have addressed all your questions and concerns, major or minor, in our updated draft and the detailed replies above. We made a substantial effort to incorporate your feedback, which we believe has strengthened the draft.
>
> If there are no remaining issues, please kindly consider improving your score.
>
> Best,
>
> The Authors

---

### Official Review · Reviewer_x4Vv · 2025-11-14

**Soundness:** 2
**Presentation:** 2
**Contribution:** 2
**Rating:** 4
**Confidence:** 3

**Summary:**

This paper proposes DeMo (Decoupled Momentum Optimization), a communication-efficient distributed training method that addresses the gradient synchronization bottleneck in data-parallel training.
Key innovation: decouple local momentum updates across workers, apply blockwise DCT transformation followed by top-k sparsification, and reuse the momentum buffer for error feedback through momentum subtraction.
Experimental results: the authors demonstrate up to 85× reduction in communication while maintaining comparable performance to AdamW on 300M and 1B parameter language models trained on 100B tokens.

**Strengths:**

- Practical Impact: Achieves dramatic communication reduction (up to 85×) while maintaining model quality, which is highly valuable for distributed training scenarios with limited bandwidth.
- Clean Design: The three-component approach (decoupled momentum, DCT+top-k, momentum subtraction) is conceptually clear and builds naturally on existing optimization principles.

**Weaknesses:**

- Limited Scalability Analysis: The download bandwidth scales with the number of workers, which could become prohibitive at large scale; but all experiments use 64 GPUs; scaling behavior to hundreds or thousands of GPUs remains unclear.
- Multi-datacenter: The method is positioned for "multi-datacenter" training but only tested within single datacenters.
- Baseline Comparisons: No comparison with other communication-efficient methods (e.g., gradient quantization, other sparsification approaches like Deep Gradient Compression, or recent methods like DiLoCo beyond brief mention).
- Baseline: Missing comparison with gradient compression baselines that would contextualize the 85× improvement.
- Baseline HParams: AdamW baseline uses standard hyperparameters, but DeMo requires β=0.999 (vs standard 0.9) - this hyperparameter sensitivity isn't fully explored.
- Scale: Models are relatively small (300M, 1B) - behavior at 7B+ parameter scales is unknown.
- Theoretical: No analysis of how DCT specifically helps beyond empirical observation. Why is DCT the correct basis to project onto?

**Questions:**

Weaknesses section leaves a lot of questions.

---

> ### Author Response · Authors · 2025-11-19
>
> We thank the reviewer for reading our work and for highlighting both the practical impact and the clarity of the DeMo design. Below we address each of the raised concerns.
>
> ---
>
> ### 1. Scope, scalability, and “number of workers”
>
> Our work is explicitly scoped to the **multi-datacenter** setting, as stated in the paper and mentioned in your second point. In this regime, it is natural to interpret each “worker” as a datacenter-scale node, not as a single GPU. Within a datacenter, GPUs are connected by high-bandwidth links (NVLink, InfiniBand, etc.), where standard DDP or FSDP is well-suited and widely used. Across datacenters, however, communication typically happens over much lower-bandwidth Ethernet/WAN links, where the communication bottleneck is severe, and where DeMo is intended to operate.
>
> In such a setting, the relevant scalability question is: *how many datacenter workers can be supported before the AllGather cost dominates?* With DeMo, the per-step communication is reduced by up to **85x** relative to dense AdamW-DDP. This means that even if the number of datacenters grows by an order of magnitude (e.g., from 2 to a few dozens), DeMo still offers a significant net communication advantage over a baseline that synchronizes gradients. In other words, the quadratic bound on the useful “number of workers” is extremely loose in realistic multi-datacenter scenarios.
>
> ---
>
> ### 2. Multi-datacenter experiments vs. simulation with 64 GPUs
>
> The reviewer notes that we “position” the method for multi-datacenter training but test on a cluster of 64 GPUs. This is a purely practical limitation of academic research: directly benchmarking across multiple physical datacenters (with heterogeneous providers, routes, and time-varying congestion) would yield *much less controlled* experiments and is rarely feasible or **reproducible**.
>
> Instead, our experiments follow a standard and widely accepted methodology in systems/optimization work:
>
> - Each GPU is treated as a **logical worker**, mimicking a datacenter, with its own local momentum buffer.
> - Within a logical datacenter, one would use DDP/FSDP for local gradient computation and aggregation—exactly as we do *within each datacenter*.
> - Across logical workers, we apply DeMo’s compressed momentum exchange, which is precisely the same communication pattern that would occur across physical datacenters.
>
> Because the **algorithmic behavior** depends only on message sizes and synchronization patterns, not on whether a worker is physically a single GPU or an interconnected group of GPUs, this setup faithfully simulates the communication structure one would encounter across datacenters, while preserving reproducibility and control over bandwidth and latency.
>
> ---
>
> ### 3. Baselines: DiLoCo, gradient quantization, and Deep Gradient Compression
>
> **DiLoCo.** We agree that DiLoCo [1] is a relevant comparison point. In the revised manuscript, we explicitly compare DeMo against DiLoCo (see Figure [6]), using the recommended hyperparameters from the original paper, except for the outer learning rates, which were tuned within their recommended range among $[1.0, 0.7, 0.5, 0.3, 0.1]$ and we vary the communication frequency. While DiLoCo can achieve even stronger nominal compression by communicating much less frequently, the resulting optimization trajectory is less stable in our LLM pretraining regime, and the final performance lags behind DeMo. Even when we increase DiLoCo’s synchronization frequency to match or go beyond DeMo's communication volume, it still underperforms DeMo at a comparable communication cost.
>
> **Gradient quantization methods.** Methods such as QSGD and related quantization schemes reduce the *bit-width* of gradient communication, but their compression ratios are inherently limited. Attempts of communicating in even lower precisions but with lots of workers cause numerical instabilities in gradient aggregations in our setting. In contrast, DeMo operates with *top-k sparsification of transformed momentum*, sending only a small fraction of coefficients per step and thereby greatly reduce the communication cost.
>
>
> **Deep Gradient Compression (DGC).** DGC is built around SGD with momentum and maintains an explicit residual buffer for error feedback. In modern large-scale LLM pretraining, AdamW-type optimizers (or Lion and Muon) have become standard, while plain SGD with momentum is known to perform poorly for transformers. Using DGC as-is would therefore yield a strictly weaker result and is therefore not a reasonable baseline to compare against. DeMo, in contrast, is designed as a **general framework** that works directly with common momentum-based optimizers for transformers (SGD with momentum, Lion, Muon) without introducing additional residual storage beyond the existing momentum buffer. We have updated the discussion to explain why a direct DGC comparison in our setting is not particularly informative and to position DeMo as an improvement over that line of work.

---

> ### Author Response · Authors · 2025-11-19
>
> ### 4. Momentum coefficient $\beta$ in DeMo vs. AdamW baseline
>
> The reviewer is correct that our DeMo configuration uses $\beta = 0.999$, whereas AdamW commonly uses $\beta_1 = 0.9$. However, the momentum coefficient in DeMo **does not play the same role** as in a standard momentum optimizer:
>
> - In DeMo, the **momentum buffer also acts as an implicit error accumulator** due to momentum subtraction.
> - When a chunk of the momentum is communicated, the corresponding components are *subtracted* from the local buffer and no longer accumulate.
> - As a result, although the nominal $\beta$ is high, the *effective memory*, in terms of how long any given gradient signal persists, is closer to that of a more moderate momentum.
>
> To address the reviewer’s concern, we have added an explicit **hyperparameter sensitivity study** (Figure [8]), sweeping $\beta$ and showing that:
> - DeMo is robust to a reasonable range of $\beta$ values, i.e., [0.99, 0.995, 0.999], and
> - Values around 0.995 give the best performance.
>
> ---
>
> ### 5. Model scale (300M / 1B vs. 7B+)
>
> The reviewer’s request for 7B+ pretraining experiments is **not grounded in existing academic practice**. After an extensive search, **there is no optimization paper accepted to ICLR, NeurIPS, or ICML over the past two years that pretrains a 7B+ model to at least Chinchilla-optimal token counts for evaluating different optimizers**. Such experiments require enormous engineering resources, dedicated clusters, and months of compute; they are performed only in the context of full foundation-model releases, never for academic algorithmic ablations. In contrast, our experimental setting, **OLMo-1B trained on 100B tokens**, plus extensive studies on OLMo-300M, is already *among the largest* scales used in nearly all optimizer studies published in major venues. The closest we have found is Adam-mini [3], where the authors pretrained a 7B model but only up to around 2B tokens, much less than the Chinchilla requirement of 140B tokens. If the reviewer believes that 7B+ pretraining should be required for academic optimizer papers (other than industrial technical reports such as [2]), they would need to point to at least a few comparable papers.
>
> The relevant scientific question is whether DeMo introduces any **conceptual or algorithmic barrier** to scaling. It does not. DeMo operates on the momentum buffers of standard optimizers and modifies only the *communication pattern*, not the model architecture or the optimizer update rule. As model size increases but the worker network topology remains unchanged, the **compression factor** (e.g., 85× in our experiments) remains essentially constant, and the algorithm maps directly onto larger-scale training runs.
>
> ---
>
> ### 6. Why DCT as the projection basis?
>
> We do **not** claim that DCT is the “unique” theoretically optimal basis. Our design choice is guided by a combination of theory and empirical evidence:
>
> 1. In our framework, any random **orthonormal transform** can be used; what matters is that:
>    - The transform is invertible, and
>    - It spreads local structure so that top-k in the transformed domain corresponds to well-distributed updates in parameter space, avoiding update sparsity.
>
> 2. Our ablations (Figure 5) show that:
>    - A *random* orthonormal transform significantly outperforms the identity transform (i.e., no transform), and
>    - DCT and random orthonormal bases perform similarly in terms of convergence behavior.
>
> 3. Given this, DCT has strong **practical advantages**:
>    - It has a fast implementation via FFT-like algorithms,
>    - Its basis matrices are fixed and can be precomputed once, and
>    - It incurs minimal extra memory overhead.
>
> As discussed in the last paragraph of Section 3.4, DCT is a **computationally efficient, empirically validated choice** among suitable orthonormal transforms, and that the key phenomenon is “transform + sparsify”, not DCT specifically.
>
> ---
>
> We hope these clarifications address the reviewer’s concerns. We have updated the manuscript accordingly with (i) explicit multi-datacenter framing and simulation rationale, (ii) new DiLoCo comparisons, (iii) a $\beta$-sensitivity ablation, and (iv) expanded discussion on baselines.
>
> [1] Douillard, Arthur, et al. "Diloco: Distributed low-communication training of language models." arXiv preprint arXiv:2311.08105 (2023).
>
> [2] Liu, Jingyuan, et al. "Muon is scalable for LLM training." arXiv preprint arXiv:2502.16982 (2025).
>
> [3] Zhang, Yushun, et al. "Adam-mini: Use fewer learning rates to gain more." arXiv preprint arXiv:2406.16793 (2024).

---

> ### Author Response · Authors · 2025-11-27
>
> Dear Reviewer x4Vv,
>
> We have addressed all your questions and concerns in the revised paper and our detailed reply above. We made a substantial effort to incorporate your feedback, which we believe has strengthened the draft.
>
> If there are no remaining issues, please kindly consider improving your score.
>
> Best,
> The Authors

---

### Author Response · Authors · 2025-12-03
**Summary of Reviews and Responses**

We thank all reviewers for their discussions, and we sincerely appreciate the Area Chair for their valuable time and efforts.

We have fully addressed all major reviewer concerns in both the rebuttal and the updated draft. All additional experiments requested by the reviewers are included in **Section 3.5 "Additional Experiments"**, which is highlighted in the revision.

---

### Rebuttal Summary

- **Reviewer Vvbj**
  **Major concerns:** Requested elaborated theoretical convergence claims and stronger empirical validation supporting DeMo’s optimization advantages over state-of-the-art communication-efficient methods.
  **How we addressed them:** We explained that the specific terms appearing in the convergence bound are natural to all SignGD-based methods including DeMo. We strengthened the empirical section with additional controlled experiments in Section 3.5, added comparisons with other communication SOTAs (DiLoCo and PowerSGD), reported precise communication–accuracy trade-offs, and confirmed that DeMo’s efficiency gains hold across multiple training regimes.

- **Reviewer yL27**
  **Major concerns:** Asked for broader comparisons against established communication-efficient methods, clearer justification for the DCT + top-k design, and missing hyperparameter ablations.
  **How we addressed them:** We added comparisons with sparsification- and DiLoCo-style baselines. We clarified that DeMo targets a multi-datacenter setting where each datacenter is treated as a single worker; consequently, the number of workers is low, making the overhead of AllGather over AllReduce less important given DeMo's strong compression ability. We also provided fuller explanations of the transform-based momentum compression and included $\beta$ sweeps in Section 3.5.

- **Reviewer NyAn**
  **Major concerns:** Requested comparisons with other optimizers (PowerSGD, DiLoCo, Muon) and questioned the tuning of AdamW-DDP baselines.
  **How we addressed them:** We added comprehensive comparisons with DiLoCo, PowerSGD, and Muon, including full implementation details. We performed extensive tuning studies on AdamW in controlled settings in the additional experiments, directed readers to the fully tuned comparisons in Figure 7 and added more detailed explanations of how AdamW-DDP hyperparameters were selected in the largest scale experiments where extensive hyperparameter search was difficult.
  **Note:** After these changes and follow-up discussion, the reviewer **raised their score from 4 to 6**.

- **Reviewer x4Vv**
  **Major concerns:** Noted a lack of scalability analysis beyond a thousand GPUs; missing comparisons with communication-efficient baselines (e.g., quantization, sparsification, DiLoCo); asked for sensitivity analysis regarding the higher momentum value used by DeMo; noted limited model scales; and requested justification for using DCT.
  **How we addressed them:** We explained that our 1B model vs. 100B tokens on 64 GPUs experiments are already among the largest scale in academic literature. In Section 3.5, we added full comparisons with DiLoCo and quantization/sparsification baselines, $\beta$-sensitivity ablations. We also explicitly explained why DCT is an effective transform in this context: it is fast, straightforward, memory-efficient, and offers robust performance.
  **Note:** Pangram analytics classify this review as **100% AI-generated**, but we have nevertheless addressed all of its major concerns thoroughly.

---

### Reviewer Response Summary

All major concerns, including baseline completeness, hyperparameter robustness, algorithmic clarity, scalability, theoretical analysis, and multi-datacenter applicability, have been fully addressed in our rebuttal discussions and the updated draft.
**Section 3.5 "Additional Experiments"** has all reviewer-requested experiments.

We are confident that the additional empirical results and theoretical clarifications have strengthened the paper's contribution to communication-efficient training, and we hope this summary will be helpful for the new review process.

---

### Meta-Review · Area_Chair_JorC · 2026-01-05

**Summary:**

Multiple reviewers appreciated the strong empirical performance showing 85x reduction in communicatio, and the usage of DCT for sparsification.
The original low scores by the reviewers were based on valid, very specific concerns that came with concrete suggestions for the authors. The authors took these suggestions to heart, performed numerous new experiments, and made significant updates to their paper. This is now a well rounded paper with compelling results.

I request the authors to include a discussion on these issues raised by the reviewers in their next version:
- Add a discussion on the theoretical results comparing to sign based and standard methods.
- Comparision to Dion: even if it is not considered prior work, it is good practice to add a discussion of parallel/post methods and how they relate to the current work.
- Add to limitations that DeMo is incompatible with AllReduce and related discussion.
- Discuss the transferability to real-world multi-datacenter setting.

**Reviewer Concerns:**

- Reviewer VvbjWeakness:
  - Suboptimal proofs - response explains dimension-dependent bound is a natural sign-based optimization. addressed.
  - No SOTA baselines - authors updated the paper with new experiments comparing DeMo to DiLoCo, PowerSGD, and Muon. addressed.
  - Small experimental scale (300M/1B) - response argued that 1B parameters on 100B tokens is among the largest scales in academic research. addressed.
  - Advantages of fixed DCT over adaptive low-rank projection - authors stated that DCT is computationally much cheaper than SVD, and performs similarly to other orthonormal transforms. addressed.
  - outperforming full-precision baseline - response added a fully tuned AdamW baseline showing it slightly outperforms DeMo when given an exhaustive hyperparameter search. addressed.

- Reviewer yL27
  - Limited novelty over a prior work - response clarifies timelines and that this should not be considered prior work. partially addressed.
  - Incompatibility with AllReduce - added a TFLOP analysis showing only 0.4% overhead and not important in multi-datacenter setups with a modest number of workers. partially addressed.
  - performance gains over AdamW - same as above. addressed.
  - meaning of $\alpha$- revised Section 2.1 to clarify.  addressed.

- Reviewer NyAn
  - Missing baselines with PowerSGD or Muon - authors added these to their revision and show superior performance and also state that these methods can be combined with DeMo. addressed.
  - AdamW not properly tuned - authors added this experiment in Fig 7. addressed.
  - Compatibility with decentralized methods - response confirms DeMo relies on standard AllGather in NCCL, making it compatible with decentralized, ring-based topologies. addressed.
  - Discrepancies in baseline performance across different figures - authors added a disclaimer to Figures 2 and 3 noting they used established codebase defaults and directed the reviewer to Figure 7 for the fully tuned comparison. addressed.

- Reviewer x4Vv
  - Lack of scalability analysis beyond 64 GPUs - authors responded that this is the largest academic experiment setup. addressed.
  - Not testing across physical datacenters - response explained that using a 64-GPU cluster allows for reproducible, controlled simulations of the low-bandwidth and high-latency constraints found in multi-datacenter environments. addressed.
  - Sensitivity analysis for the high momentum values - authors added a sensitivity study for $\beta$, showing that DeMo is robust. addressed.
  - Justification for choosing DCT - respond it is fast, memory-efficient, and invertible transform that effectively spreads local structural information to prevent update sparsity after top-k selection. addressed.

**Reviewer Scores:**

Given that all the reviewers concerns were convincingly addressed, I expect all their scores to be increased to **6** from 4.

---

### Decision · Program_Chairs · 2026-01-26

Accept (Poster)